# Sensitivity analysis of a built environment exposed to the synthetic monophasic viscous debris flow impacts with 3-D numerical simulations

Xun Huang[1,2], Zhijian Zhang[1], Guoping Xiang[3,4]

[1]Geography and Tourism College, Chongqing Normal University, Chongqing, 401331, China
[2]Chongqing Key Laboratory of Surface Process and Environment Remote Sensing in the Three Gorges Reservoir Area, Chongqing Normal University, Chongqing, 401331, China
[3]405 Geological Brigade of Sichuan Bureau of Geology & Mineral Resources, Dujiangyan, 611830, China
[4]State Key Laboratory of Geohazard Prevention and Geoenvironment Protection, Chengdu University of Technology, Chengdu, 610059, China

*Correspondence to*: Xun Huang (huangxun@cqnu.edu.cn)

**Abstract.** The characteristics of exposed built environments have a significant effect on debris flow impacts on buildings, but knowledge about their interactions is still limited. This paper presents a sensitivity analysis on the peak impact forces on a whole building resulting from the built environment parameters, including the orientation, opening scale of the target building, and azimuthal angle and distance of surrounding buildings. The impact forces were obtained from the monophasic viscous debris flow with a synthetic and simplified hydrograph using the FLOW-3D model, a computational fluid dynamics approach, verified through the physical modeling results. The results show that the surrounding buildings' properties have significant roles in determining the peak impact forces. A shielding effect or canalization effect, which reduce or increase impact forces, respectively, can be produced by changing the azimuth angle. A deflection wall for building protection is recommended according to the shielding effect. A narrowed flow path, determined by both the azimuth angle and distance, has a significant effect on the variation in impact forces. In addition, it is concluded that a splitting wedge should be designed following a criterion of avoiding the highest flow depth - the maximum approaching angle - appearing near the longest wall element. The protruding parts caused by changing the building's orientation contribute to increasing impact loads within a shielding area. A limited opening scale effect is observed on the whole building if there is sufficient time for material intrusion. The insights gained contribute to a better understanding of building vulnerability indicators and local migration design against debris flow hazard.

## 1 Introduction

In mountain environments, buildings are the elements of greatest concern with regard to debris flow hazard risks (Zeng et al., 2015; Fuchs et al., 2019; Luo et al., 2020). Compared to strong structures such as railway bridges, common residential buildings are more easily damaged by debris flows (Hu et al., 2012; Huang and Tang, 2014). Furthermore, the damage to

buildings has been demonstrated to contribute greatly to casualties and property loss based on a large number of catastrophic debris flow events (Tang et al., 2011; Zhang et al., 2018; Chen et al., 2021). In recent decades, the understanding, assessment and eventual reduction in building exposure and vulnerability have been brought into focus in mountain hazard mitigation (Holub and Fuchs, 2009; Fuchs et al., 2015, 2017).

In the classical S-shaped vulnerability curves, a quantitative assessment approach based on interactions between process intensities and building damage, considerable ranges in the loss ratio were found for moderate process intensities (Fuchs et al., 2012). This is in good agreement with the fact that two buildings located exactly at the same place, despite experiencing the same process intensity, do not always suffer the same degree of loss (Papathoma-Köhle, 2016; Papathoma-Köhle et al., 2017). Process intensity is not decisive in terms of building damage, and some building characteristics of the building itself

and its surroundings play critical roles in the degree of damage induced by debris flow. This understanding has also been confirmed by the spatial distribution of building damage ratios in the debris flow torrents of the Austrian Alps (Fuchs et al., 2012).

Currently, the characteristics of an exposed building, including the orientation, openings and surrounding environments, have been reported to be important to the impact forces of torrential hazards (Jakob et al., 2012; Sturm et al., 2018a). As far

as building orientation is concerned, it is commonly recommended to individually analyze wall elements of different orientations. It is widely accepted that walls with faces perpendicular to the stream are generally exposed to higher dynamic pressures due to the larger effective contact area (Mead et al., 2017; Manawasekara et al., 2016). However, limited studies to date have examined the impact performance of a whole building. In terms of building openings such as windows, doors and light shafts, it is well established that the impact forces on the building envelope tend to decrease once the flow penetrates

openings (Mazzorana et al., 2014; Gems et al., 2016). However, this process is regarded to be at the expense of greater building loss due to the higher impact forces on interior walls and greater indoor property losses (Totschnig et al., 2011; Mead et al., 2017). Recent developments regarding the surroundings of built environments have attracted much attention. The existence of surrounding buildings definitely reduces the impact forces on a particular building due to the deflection of the flow and the shielding of the element at risk; however, this may also increase the impact forces by redirecting or

canalizing the flow and forcing the flow to impact the building (Gao et al., 2017; Sturm et al., 2018b). Therefore, it remains difficult to make a general statement regarding whether the effect of surrounding buildings is negative or positive (Sturm et al., 2018a).

Although considerable efforts have been devoted to the relationships between building characteristics and debris flow impacts, only some simple trends, rather than quantitative results about the effects of factors, have been determined.

Knowledge regarding the interactions between the built environment parameters and impact loads is still limited, and considerable research gaps still exist regarding (1) the identification of the built environment parameters with the greatest influence on debris flow impacts; (2) detailed explanations of each built environment factor; and (3) the interrelations between these parameters.

Because field measurement of debris flow impacts is nearly impossible, laboratory experiments and numerical modeling are regarded as feasible alternatives for capturing interactions, and they may provide insights into the impact of debris flows in the interior and against the exterior of buildings (Gems et al., 2016; Papathoma-Köhle, 2016). In this study, a sensitivity analysis was conducted based on 3-D numerical simulation, and the effects of various built environment factors on the impact forces of debris flows were quantitatively analyzed and compared. Debris flow numerical simulations were conducted using the FLOW-3D model, a commercial Computational Fluid Dynamics (CFD) program. Kim et al. (2021) conducted a sensitivity analysis of five parameters for fine sediment trapping and energy reduction with a debris flow slit-type barrier via a FLOW-3D numerical model and metamodels. These kinds of methods offer opportunities for performing a sensitivity analysis with limited data (Kim et al., 2019, Kim et al., 2021). These results can be applied to determine the indicators and to improve weightings for reliable building vulnerability assessment and to enhance the knowledge about built environment improvement and local migration design.

In the following, the reliability of the numerical model was first confirmed by comparison with a physical dam-break experiment. A sensitivity analysis was then performed using metamodels and global sensitivity analysis (GSA). The built environment parameters considered in this study were the orientation (*Or*) and opening scale (*Op*) of the target building and the azimuthal angle (*A*) and distance (*D*) of the surrounding buildings. Finally, the effects of each parameter and their interactions on the peak impact forces of the overall building were explained in detail.

## 2 Numerical modeling of debris flow

### 2.1 Model description

The identification of complex geometry and three-dimensional flow tracking are the key steps in the process-response simulation between buildings and debris flows. However, the current numerical codes used for debris flow simulations consider the flow depth to be small relative to the tangential length scale and simplify this factor to represent shallow water flow. They have only second-order accuracy in space, as the effects of complex three-dimensional topography and the vertical mobility of debris flows are not considered (Zhang et al., 2021b). To avoid the above-mentioned limitations, it is necessary to use an efficient 3-D numerical approach to accurately capture the debris flow behaviors considering the influences of building geometry.

FLOW-3D, a three-dimensional finite-volume-based CFD model, is considered one of the most efficient tools for predicting hydraulic phenomena with strong turbulent components and inconsistent free water surfaces. FLOW-3D was designed to address the Reynolds-Averaged Navier-Stokes equations (RANS) (Jones and Launder, 1972), implementing the Volume of Fluid (VOF) methods (Hirt and Nichols, 1981) and Fractional Area/Volume Obstacle Representation (FAVOR) (Hirt and Sicilian, 1985). The advanced Tru-VOF method can be used to precisely track the three-dimensional transient free fluid surface. Its unique FAVOR mesh processing technology can define independent and complex geometry within the structured mesh and avoid the shortcomings of the traditional finite difference method in complex boundary fitting (Zhang et al.,

2021b). The FAVOR processor can generate area fractions for each cell face in the grid by determining which corners of the face are inside of a defined geometry, and incorporate geometry effects into the governing equations. As a result of a robust capacity to deal with the data in both the fluid and solid phases, the FLOW-3D code is considered to be appropriate for analyzing the interactions between debris flows and exposed buildings.

In this study, the renormalized group (RNG) model-based k-ε turbulence model and the general moving objects (GMO) model are applied to build fluid-solid coupled model of the debris flow impact. The RNG k-ε model is a modification of the standard k-ε model, which takes the turbulent vortex into account and provides an analytic formula for Prandtl number, as well as an analytic formula for low Reynolds number flow viscosity (Franco et al., 2021). These features make the RNG model more reliable and accurate for a broader flow than the standard k-ε model (Yin et al. 2015). In recent years, the RNG

model has been applied in simulations of landslide surges (Yin et al., 2015; Hu et al., 2020), the entrainment effects of debris avalanches (Hu et al., 2019), dam-break floods (Zhuang et al. 2020), and the runout characteristics of debris flows (Zhang et al., 2021b), with good results.

The turbulent kinetic energy and the turbulence dissipation balance equations of RNG k-ε model in FLOW-3D are as following:

$$\frac{\partial k_T}{\partial t} + \frac{1}{V_F}\left\{uA_x\frac{\partial k_T}{\partial x} + vA_y\frac{\partial k_T}{\partial y} + wA_z\frac{\partial k_T}{\partial z}\right\} = P_T + G_T + Diff_{k_T} - \varepsilon_T \tag{1}$$

$$\frac{\partial \varepsilon_T}{\partial t} + \frac{1}{V_F}\left\{uA_x\frac{\partial \varepsilon_T}{\partial x} + vA_yR\frac{\partial \varepsilon_T}{\partial y} + wA_z\frac{\partial \varepsilon_T}{\partial z}\right\} = \frac{CDIS1 \cdot \varepsilon_T}{k_T}(P_T + CDIS3 \cdot G_T) + Diff_\varepsilon - CDIS2\frac{\varepsilon_T^2}{k_T} \tag{2}$$

where $k_T$ is the turbulent kinetic energy, $V_F$ is the fractional volume open to flow, $A_x$, $A_y$ and $A_z$ are the fractional area open to flow in the x, y and z directions, respectively. $P_T$ is the turbulent kinetic energy production term, $G_T$ is the buoyancy production term, $Diff$ is the diffusion term, and $\varepsilon_T$ is the turbulence dissipation term. In the RNG model of FLOW-3D, $CDIS1$

and $CDIS3$ are dimensionless user-adjustable parameters that have defaults of 1.42 and 0.2, respectively, and $CDIS2$ is determined from $k_T$ and $P_T$ (Flow Science, Inc., 2014).

In this study, a series of debris flow simulations based on the RNG model with various building characteristics were executed. From the characteristics of RNG k-ε model, the type of debris flow involved in this study was determined as mudflow or viscous debris flow, in which a single-phase fluid was assumed and solid particles were treated as suspension

and mixed with the fluid phase well. The division between solid and fluid was assumed to be difficult, therefore, the granular deposition was not considered in this study.

With the help of GMO model in FLOW-3D, the combined hydraulic force due to normal pressure and shear stress can be calculated in the space system. The normal pressure and shear force of an impacted object in x, y and z directions can be gained at each time step. Due to the complex geometry and variable built environments, the impacted elements of target

building were changing in the different scenarios. In this study, therefore, the target building was treated as a whole bearing structure to keep consistency of analysis. All over the grids covering building surface would be calculated when contacting with the flow. The GMO model can simulate the rigid body motion, which is either user prescribed or dynamically coupled with fluid flow. In this study, the target building and surrounding buildings were all prescribed to be the fixed and non-

deformable rigid models. The raw impact force data was collected at interval of 0.001 s from the numerical code. To reduce the uncertainty, a simple data noise reduction approach, that the peak impact forces were obtained from the average values over 10 points in the timeline (0.01 s), was executed (Song et al., 2021).

## 2.2 Model validation

The interaction between a dam-break and the structure has become a classic benchmark for the validation of fluid-structure interaction (Liu et al., 2021). The accuracy of the model will be validated by means of the experimental setup previously used in Gomez-Gesteira and Dalrymple (2004). This experiment has been referred as a "bore in a box", where it was a dam-break and structure-impact problem confined within a rectangular box. The geometric dimensions of the experimental model are shown in Fig. 1. The rectangular tank is 1.60 m long, 0.61 m wide and 0.75 m high. The tank is considered to be smooth surface, and its surface roughness ($k$) is set 0 m (Liu et al., 2021). The volume of water initially contained behind a thin gate at one end of the box is 0.4 m long, 0.61 m wide and 0.3 m high. An initial layer of water (approximately 1 cm deep) existed on the bottom of the tank. The obstacle, which is 0.12 m $\times$ 0.12 m $\times$ 0.75 m in size, is placed 0.5 m downstream of the gate and 0.24 m from the nearest sidewall of the tank. The surface roughness of the obstacle is not determined, $k_o$ (0, 0.001, 0.002, or 0.003 m), therefore, is selected for sensitivity analysis to determine if this parameter could reasonably reflect the macro mechanical behaviors of the dam-break test. The time history of the impact force on the structure was measured with a load cell.

In the numerical simulation, the analysis domain was discretized into a grid with a cell size of 0.01 m, which was equal to a cube of 0.01 m side in 3-D model. The fluid properties were set to be the density of 1000 kg m$^{-3}$ and viscosity of 0.001 Pa·s. The motion of fluid was computed by means of RNG k-ε model in FLOW-3D. The obstacle and gate were controlled by the GMO module, specifically this obstacle was set as a fixed and non-deformable rigid body, and the gate was prescribed to be lifted 0.3 m along the z$^+$ direction. The time history of impact forces and the corresponding dynamic processes were selected to validate the accuracy of the numerical simulation. The direction of the force was considered positive when exerted in the y$^+$ direction.

Fig. 2a shows the general agreement of numerical forces obtained by means of the RNG and GMO coupled model with experimental data, particularly the positions of both peaks, which correspond to the wave hitting the front and the back of the structure and were reasonably reproduced by the numerical model. This indicates that the impact force is very low sensitivity to $k_o$, this study does not pay too much attention to the value of this parameter, and the surface roughness ($k$) of all the impacted object in numerical simulation is set 0 m. Fig. 2b shows the evolution of the wave generated by the dam-break and the initial layer of water on the bottom. At t = 0.32 s, the wave is colliding with the front of the obstacle. At t = 0.58 s, the wave is wrapping around the structure, colliding together and continues moving toward the tank wall. At t = 1.44 s, the reflected wave is hitting the back of the obstacle.

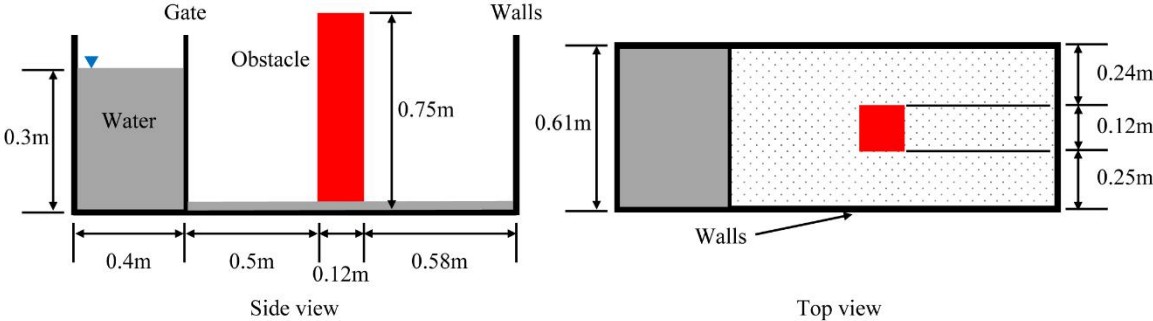

**Figure 1. The geometric dimensions of the experimental dam-break model**

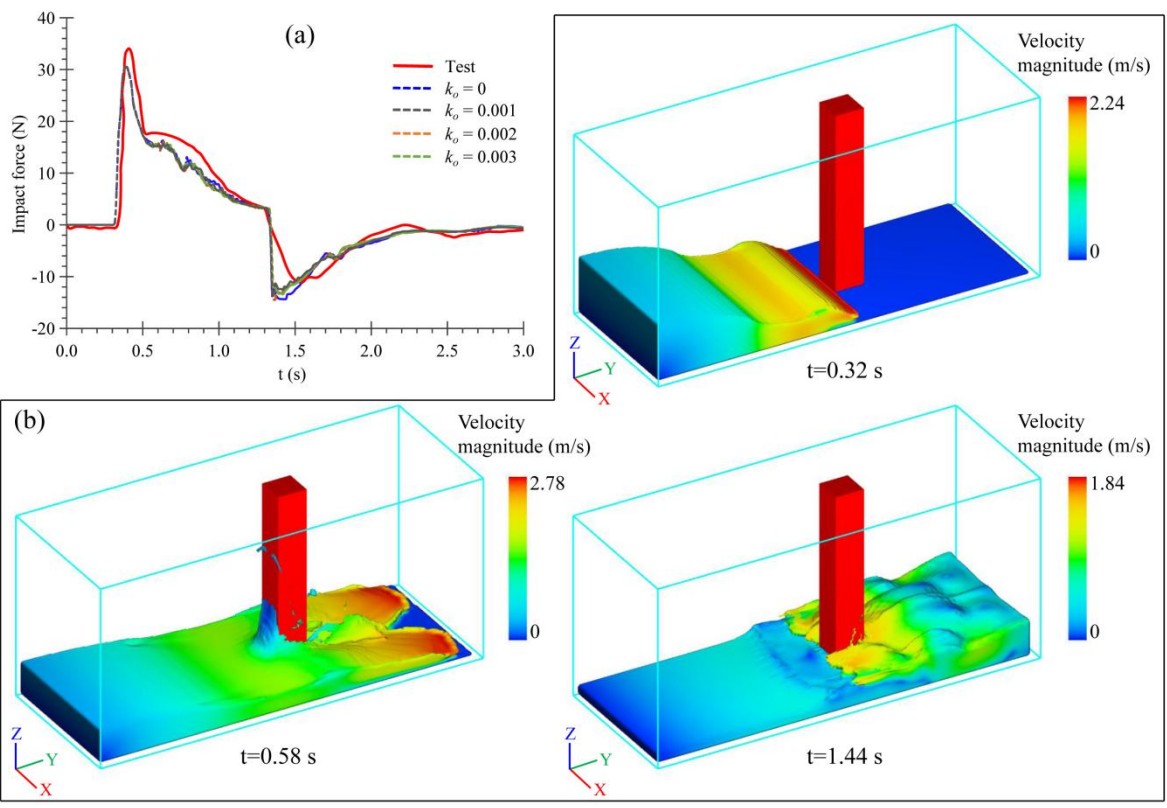

Figure 2. The dam-break simulation (a) comparison between numerical (dotted lines) and experimental values (red line) of the force exerted on the structure; (b) wave evolution (t = 0.32 s) the wave colliding with the front of the obstacle; (t = 0.58 s) the wave wrapping around the structure, colliding together and continues moving toward the tank wall; (t = 1.44 s) the reflected wave hitting the back of the obstacle.

## 2.3 Numerical model set up

As shown in Fig. 3, a depositional fan model with a length of 120 m, width of 120 m and monogradient of 5° was numerically constructed. This deposition fan was treated as a rigid bed model, therefore, the bed material scour would not happen in this study. To sum up, the bed variation induced by sediment transport processes, including sediment deposition

and bed scour, was not considered. The surface roughness ($k$) of 0.05 m was set, meaning that the deposition fan surface was roughened with 5 cm diameter particles, for the representation of the natural environment surrounding mountainous buildings.

The realistic debris flow inflow discharge and duration are essential for a truthful analysis of impact forces. A 3-D numerical simulation with a realistic hydrograph, however, needs a large number of computer memory and processing time. It is much hardly acceptable for the sensitivity analysis, in which the sufficient experimental groups are required. In this study, a synthetic and simplified inflow hydrograph was set at the top center of the deposition fan. Specifically, the debris flow discharge was fixed as 500 m$^3$ s$^{-1}$ which can be of some interest for a large magnitude of debris flow, and the inflow duration was limited to 5 s. The inflow cross section was rectangular, with a channel base width of 20 m, flow depth of 2.5 m, inclination angle of 5° and an initial velocity of 10 m s$^{-1}$. In consideration of the distance between inflow cross section and target building, a computation time of 15 s, triple of the inflow duration, was set for the FLOW-3D modelling, to ensure all the debris flow can flow through the target building. The peak impact force was treated as the maximum value in a relatively complete impact process. In order to compare with some realistic building damage cases, for example the Qipan gully and Zhouqu debris flows in the west of China, the rheological properties of numerical model was set as the viscous debris flow. Therefore, the debris flow density was set as 2000 kg m$^{-3}$ and viscosity was empirically 1.0 Pa·s (Takahashi, 2007).

The target and surrounding building models, with a length of 15 m, width of 10 m, height of 6 m (equal to 2 floors) and wall thickness of 0.35 m, were designed in accordance with representative buildings in the mountainous areas of the western China. To maintain a balance between the computational accuracy and time cost, the whole computation domain was discretized at intervals of 0.5 m. The planar center of the target building was set 80 m downstream of the debris flow inflow, as shown in Fig. 3. The embedded domain of target building should be refined further following two principles: (1) its cell size must be less than the wall thickness of 0.35 m. This is because it is possible that the wall geometry may intersect a cell face more than once in case of building rotating, and the corresponding cell edge is assumed to be either fully inside the object or fully outside, some lacks of wall surface will be produced. (2) the boundaries of the embedded and external meshes must be overlapped for the computation stability, that is the external cell size is a multiple of embedded cell size. To sum up, the cell size of target building domain was determined as 0.25 m. The total number of computational cells was 1,645,600 and the computer memory of a single simulation was about 20 GB.

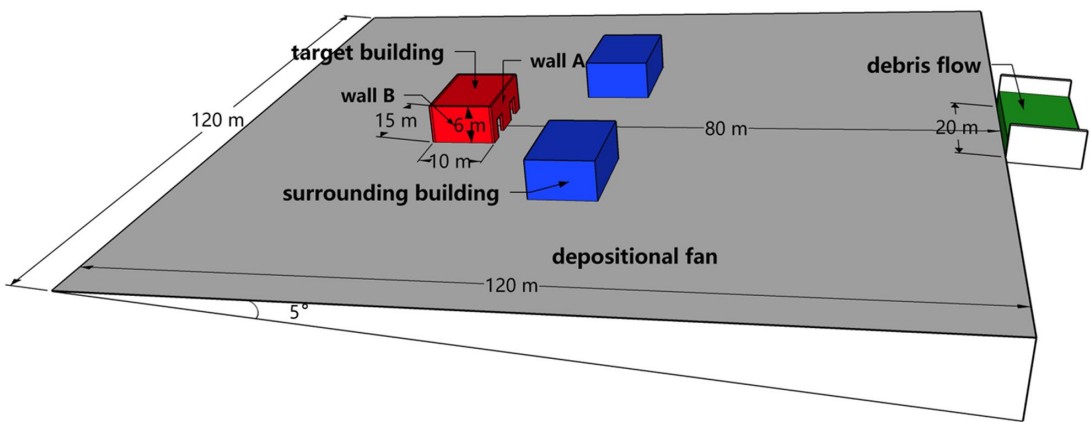

**Figure 3. Overview on the components of three-dimensional deposition fan model for debris flow impact simulation on buildings.**

Because the sensitivity analysis requires extensive simulation results to consider the combinations of all the factors, the representative factors that could be adjusted in the FLOW-3D simulation were chosen (Kim et al., 2021). The orientation (*Or*) and opening scale (*Op*) of the target building and the azimuthal angle (*A*) and distance (*D*) of the surrounding buildings with respect to the target building were considered to be the key built environment parameters for debris flow impact loads.

In this study, the orientation (*Or*) of the target building was defined as the angle between the building's long axis and the debris flow's main flow path, as shown in red and green in Fig. 4a and ranged from 0° to 90°. An orientation of 0° meant that the long axis of the building was parallel to the debris flow path, and the perpendicular case was represented by an orientation of 90°. As far as building openings are concerned, it is well known that several features of openings are of great importance in terms of building damage, such as which wall they are located on and their size, height and structure (Gems et al., 2016; Faisal et al., 2018; Papathoma-Köhle et al., 2019). However, it remains challenging to analyze these parameters with a single model. In this study, the size of the opening was only selected to be analyzed to reduce computing costs. As shown in Fig. 4b, two symmetrical openings with a constant height of 3 m were placed in wall A. Therefore, the opening scale (*Op*) was defined as the proportion of the total opening width (double *w*) to the length of wall A.

The azimuthal angle (*A*) of the surrounding buildings was defined as the angle between the line of the geometric center of buildings (e.g., line B-B' in Fig. 4c) and the main flow path (green line in Fig. 4c). There were no surrounding buildings downstream of the target building. Except for the scenario of azimuthal angle 0°, two surrounding buildings were placed symmetrically on both sides of the target building. The distance (*D*) between the surrounding building and target building was defined as the straight-line distance between two points located on the overlap between the line of the buildings' geometric centers and building envelope, as shown by the orange line C-C' in Fig. 4c.

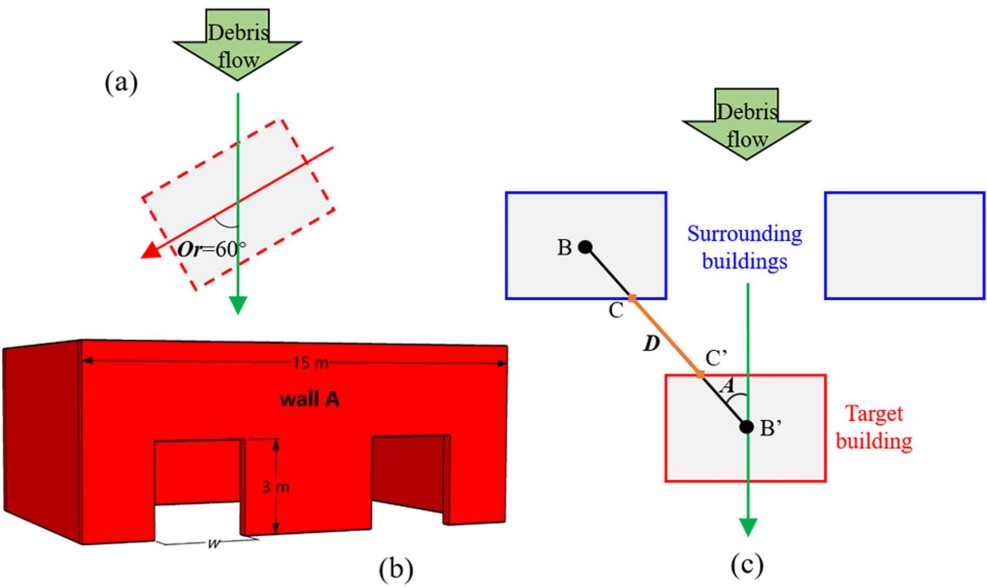

**Figure 4. The schematic drawings of the target building and surrounding buildings: (a) orientation of target building (*Or*); (b) opening scale of the target building (*Op*); and (c) azimuth angle (*A*) and distance (*D*) of surrounding building with respect to the target building.**

## 3 Sensitivity analysis

### 3.1 Metamodel modeling

Abundant simulation results are required for assessing the effect of built environment parameters on the debris flow impact force. Due to the considerable time consumed and computational cost, a mathematical metamodel was constructed using a small fraction of the simulation results. A metamodel, referred to as a surrogate model, is a 'model of a model', a simplified model of an actual model using mathematical construction. Numerous accurate simulation results can be generated based on the metamodel relation or algorithm between input and output (Booker et al., 1999; Hoffman et al., 2003). Of the various metamodel modeling methods, the Kriging model - or Gaussian process (GP) - is considered to be the most suitable for the unbiased prediction of a deterministic model and is also fitted to simulation I/O data obtained for global experimental areas (Kleijnen, 2016). In this study, the GP was selected and executed through JMP® Pro 16.0.0, a commercial statistical software designed by SAS Institute Inc.

In this study, the metamodels representing the objective function were created using the 160 samples shown in Table 2. In the metamodel modeling, the variation range of the orientation (*Or*) was from 0° to 90°, the opening scale (*Op*) was from 0 to 0.8, the azimuthal angle (*A*) was 0° to 90° and the distance (*D*) was from 5 m to 30 m. The coefficient of determination ($R^2$) of regression analysis in the GP model was 0.88. At this accuracy, a total of 10,000 new simulation results were obtained from random values in the specified ranges of the four input variables for use in the subsequent sensitivity analysis.

**Table 2. Simulation conditions.**

| Group | Number of cases | Target building's properties | | Surrounding buildings' properties | |
|---|---|---|---|---|---|
| | | Orientation (*Or*, °) | Opening scale (*Op*) | Azimuth angle (*A*, °) | Distance (*D*, m) |
| A1[a] | 5 | 0, 30, 45, 60, 90 | 0.4 | 45 | 15 |
| A2 | 25 | 0 | 0.4 | 0, 30, 45, 60, 90 | 5, 10, 15, 20, 30 |
| A3 | 25 | 30 | 0.4 | 0, 30, 45, 60, 90 | 5, 10, 15, 20, 30 |
| A4 | 25 | 60 | 0.4 | 0, 30, 45, 60, 90 | 5, 10, 15, 20, 30 |
| A5 | 5 | 90 | 0, 0.2, 0.4, 0.6, 0.8 | 45 | 15 |
| A6 | 25 | 90 | 0.2 | 0, 30, 45, 60, 90 | 5, 10, 15, 20, 30 |
| A7 | 25 | 90 | 0.4 | 0, 30, 45, 60, 90 | 5, 10, 15, 20, 30 |
| A8 | 25 | 90 | 0.6 | 0, 30, 45, 60, 90 | 5, 10, 15, 20, 30 |
| B1[b] | 5 | 0, 30, 45, 60, 90 | 0 | null | null |
| B2 | 3 | 45 | 0.4 | 0 | 5, 10, 15 |
| B3 | 5 | 90 | 0, 0.2, 0.4, 0.6, 0.8 | null | null |

[a] Simulations group 'A' were designed for the metamodel modeling.
[b] Simulations group 'B' were designed for the detailed interpretations of sensitivity analysis results.

## 3.2 Sensitivity analysis

Sensitivity analysis aims to understand the relative importance of uncertain input variables to the model response (Zhang et al., 2021a). As opposed to local sensitivity analysis, GSA can be performed by an all-at-a-time method, where output variations are induced by varying all input factors simultaneously, and thus, the sensitivity of each factor considers the direct influence of the factor as well as the joint influence caused by the factor interactions (Kim et al., 2019). GSA allows a ranking among the input parameters to be established according to their influence on the variability of the output. In this study, GSA was conducted to simultaneously consider both the main and interaction effects of input parameters on debris flow impact. A variance-based GSA (VBSA), also referred to as Sobol's indices, is usually recommended. This method is applicable over the whole space of random input data and can also deal with nonlinear responses and measure the effect of interactions in nonadditive systems (Saltelli et al., 2010). The basic principle of Sobol's indices is that the variance of model output is decomposed into fractions within a probabilistic framework that can be attributed to inputs and sets of inputs (Sobol, 1993):

$$V = \text{Var}[f(x)] = \sum_{i=1}^{d} V_i + \sum_{1 \leq i < j \leq d}^{d} V_{ij} + \cdots + V_{1,2,\ldots,d} \tag{3}$$

where the partial variances are calculated as follows:

$$V_{i_1,\ldots,\,i_s} = \int f_{i_1,\ldots,\,i_s}^2 \left(x_{i_1}, \ldots, x_{i_s}\right) p\left(x_{i_1}, \ldots, x_{i_s}\right) dx_{i_1}, \ldots, x_{i_s}, \ s = 1, \ldots, d \tag{4}$$

Sobol's indices are defined as the relative contribution of the partial variances to the total variance following the decomposition in Eq. (1)

$$S_{i_1,\ldots,\,i_s} = \frac{V_{i_1,\ldots,\,i_s}}{V} = \frac{V_{i_1,\ldots,\,i_s}}{\sum_{i=1}^{d} V_i + \sum_{1 \leq i < j \leq d}^{d} V_{ij} + \cdots + V_{1,2,\ldots,d}} \tag{5}$$

such that:

$$\sum_{i=1}^{d} S_i + \sum_{1 \leq i < j \leq d} S_{ij} + \ldots + S_{1,2,\ldots,d} = 1 \qquad (6)$$

where the index $S_i$ measures the separate contribution of each variable $x_i$ to the output variance without interaction with any other inputs; hence, $S_i$ is commonly referred to as the first-order effect index or the main effect index. The higher-order indices in Eq. (4) measure the interactive contribution to the total variance. Using $S_i$, $S_{ij}$ and higher-order indices, we can therefore infer the impacts of each input variable and the interaction of variables on the output variance (Zhang et al., 2021a).

In this study, only the second-order effect index, reflecting the interaction between every two factors, was considered. The total contribution of variable $i$ is as follows:

$$S_i^T = \sum_{\{i\} \subset \{i_1,\ldots,i_s\}} \frac{V_{i_1,\ldots,i_s}}{V} \qquad (7)$$

which measures the contributions of variable $x_i$ and its interactions to the output variance. If the input $x_i$ has a $S_i^T$=0.5, then it contributes 50% of the overall variance of output. In Sobol sensitivity analysis, the input factors with a sensitivity index

below 0.01 are usually considered noninfluential to the output (Sarrazin et al., 2016). Unlike the first-order indices,

$$\sum_{i=1}^{d} S_i^T \geq 1 \qquad (8)$$

because the interaction effect between, for example, $x_i$ and $x_j$ is included in both $S_i^T$ and $S_j^T$. The sum of $S_i^T$ is equal to 1 if and only if the model is purely additive without any interaction effects.

In this study, Sobol's global sensitivity indices were calculated using the SobolGSA model, a general purpose GUI-driven

GSA software developed by Kucherenko and Zaccheus (https://www.imperial.ac.uk/process-systems-engineering/research/free-software/sobolgsa-software/). SobolGSA evaluates the effect of a factor while all other factors are varied as well, and thus, it accounts for interactions between variables and does not depend on the choice of a nominal point like local sensitivity analysis methods. The set of available GSA techniques includes screening method- (the Morris measure), variance- (Sobol' indices, FAST) and derivative-based sensitivity measures. All techniques implemented in

SobolGSA make use of either quasi-Monte Carlo sampling based on Sobol sequences or standard Monte Carlo sampling (Sobol et al., 2011; Kucherenko et al., 2015).

## 4 Results and discussion

### 4.1 Results of global sensitivity analysis

Global sensitivity indices and total sensitivity indices for debris flow impacts are listed in Table 3, and the main and total

285 effects of each parameter are expressed in Fig. 5. From the main effect indices, the peak impact force is most sensitive to the azimuth angle, with a maximum value of 0.6303, which represents 63.03% of the overall variance in the debris flow peak impact loads. The most influential second-order effect index, with a value of 0.1842, was obtained for the interaction between the azimuth angle and the distance of the surrounding buildings. This result also highlights the importance of the surrounding buildings' azimuth angle to the debris flow impact over the distance's main effect index of 0.0095. The sums of

all main effect and second-order effect indices are 0.7648 and 0.2352, respectively, indicating that single built environment parameters have a more significant effect on the debris flow impact on a building.

**Table 3. Global sensitivity indices and total sensitivity indices.**

| Input variables | *Or* | *Op* | *A* | *D* |
|---|---|---|---|---|
| *Or* | 0.1200 | 0.0026 | 0.0346 | 0.0077 |
| *Op* | 0.0026 | 0.0050 | 0.0040 | 0.0021 |
| *A* | 0.0346 | 0.0040 | **0.6303** | **0.1842** |
| *D* | 0.0077 | 0.0021 | 0.1842 | 0.0095 |
| Total effects | 0.1649 | 0.0137 | 0.8531 | 0.2035 |

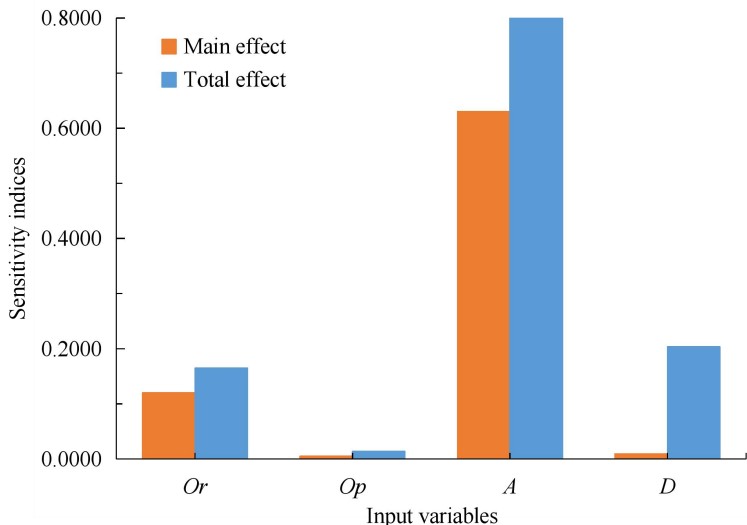

**Figure 5. Main and total effects of parameters for debris flow peak impact forces.**

From the total effect indices, on the other hand, the importance of the built environment parameters to the debris flow impact responses are ranked as follows: azimuth angle (*A*) > distance (*D*) > orientation (*Or*) > opening scale (*Op*), and their total effect indices are 0.8531, 0.2035, 0.1649 and 0.0137, respectively. The sum of the main effects of the target building's properties (*Or* + *Op*) is 0.1786, and that of the surrounding buildings' properties (*A* + *D*) is 1.0566, indicating that the surrounding buildings' properties are more significant than the target building's properties on the peak impact forces.

Finally, it is concluded that the azimuth angle and distance of the surrounding buildings and the target building's orientation are the key factors and must be carefully considered in the assessment of building vulnerability to debris flow impacts. It is highly recommended to further study the effect of the surrounding buildings' azimuth angles. Although the scale of building openings indeed has a nonsignificant effect on debris flow impacts in this study, the other features of building openings, such as their location, height and structure, should be discussed in more depth.

## 4.2 Effect of the surrounding buildings' azimuthal angles

As shown in Fig. 6, the peak impact forces of the debris flow change with the increasing azimuth angles in the simulation scenarios at an orientation of 90°, an opening scale of 0.4 and a distance of 5 m (*Or*90-*Op*0.4-*A*x-*D*5, where x represents a variable). The variations in the peak impact forces are calculated with the background value of 3717 kN from the scenario with an orientation of 90°, opening scale of 0.4 and no surrounding buildings (*Or*90-*Op*0.4-*A*null-*D*null, where null means the value is not relevant). The azimuth angles have different kinds of effects on debris flow impacts, as shown in the different colored zones in Fig. 6:

(1) Shielding effect: In the cases of azimuth angles of 0° (*Or*90-*Op*0.4-*A*0-*D*5) and 30° (*Or*90-*Op*0.4-*A*30-*D*5), the peak impact forces are only 796 and 1193 kN, and the corresponding variations ($\Delta F$) are -78.58% and -67.90%, respectively, which are located in the blue shielding effect region ($\Delta F < -10\%$) in Fig. 6. It is indicated that the target building is protected effectively in a shielding area produced by the surrounding buildings. This result is consistent with previous case studies wherein some representative catastrophic debris flow events were observed. In the Zhouqu debris flow event shown in Fig. 7, which occurred in Gansu Province of northwestern China, on 7 August 2010, building B, next to the extensively damaged building A, suffered no damage apart from its first story being buried (Hu et al., 2012). Similarly, in the Qipan gully debris flow case (Zeng et al., 2015), which occurred in Wenchuan city of southwestern China during rainstorms on 11 July 2013, building B, shielding protected from the completely damaged building A, was exposed to only slight damage, as shown in region I of Fig. 8. As shown in Fig. 9, a deflection wall was designed following the principles of the shielding effect and can be repeatedly used to protect an entire building ensemble from gravitational mass movements, such as the snow avalanches, in the mountain areas of Austria (Holub et al., 2012). This design of local protection can provide an effective reference for debris flow mitigation.

(2) Canalization effect: As reported by Sturm et al. (2018a) through flume tests, the existence of surrounding buildings may narrow the flow path and even redirect debris flows, leading to an increasing process intensity toward other buildings (Gao et al., 2017). In these numerical simulations, the azimuth angle of 45° (*Or*90-*Op*0.4-*A*45-*D*5) has the most significant canalization effect on the target building and produces the steepest rise in the peak impact force, with 29.14% growth to a maximum value of 4800 kN, as shown in the orange region ($\Delta F > 10\%$) of Fig. 6. As shown in region IV of Fig. 8, building I was exposed to extensive damage in the Qipan gully debris flow partly because of the canalization effect induced by surrounding building J.

(3) Noneffect: The peak impact forces in cases of azimuth angles of 60° (*Or*90-*Op*0.4-*A*60-*D*5) and 90° (*Or*90-*Op*0.4-*A*90-*D*5) are 3584 and 3725 kN, respectively, close to the background value (3717 kN), and the corresponding variations are also close to 0, located in the green region ($-10\% \leq \Delta F \leq 10\%$) of Fig. 6. This indicates that there are very small or even negligible effects on the impact loads of the target building. It is important to highlight that the shielding effect has the greatest influence on debris flow impact loads because this effect yields the maximum variation in peak impact forces.

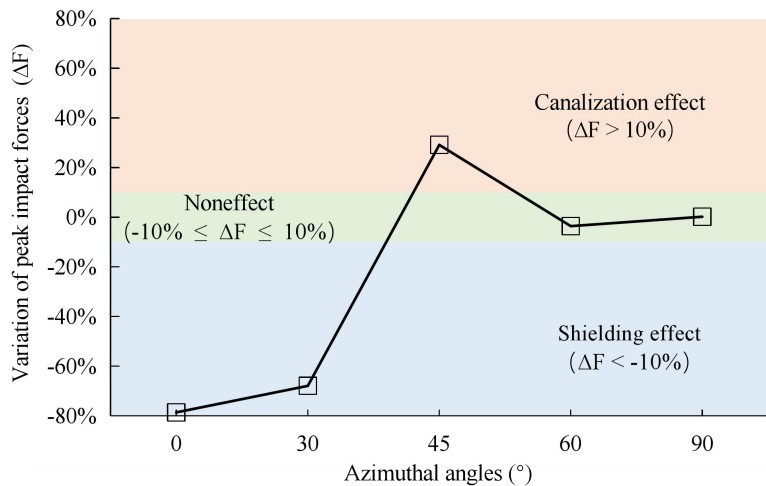

**Figure 6. Variations of peak impact forces with increasing azimuth angles in scenarios of orientation 90°, opening scale 0.4 and distance 5 m (*Or*90-*Op*0.4-*A*x-*D*5).**

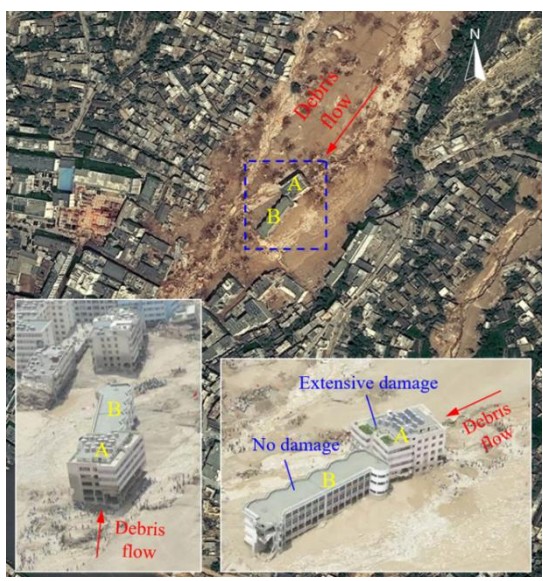

**Figure 7. Distribution of damaged-buildings in Zhouqu debris flow event. Orthophoto map is from Digital Globe's WorldView-2 satellite on 22 October 2010 published by NASA (ref: https://earthobservatory.nasa.gov/images/45329/landslide-in-zhouqu-china);**
**Aerial photo is referred from website of http://slide.news.sina.com.cn/c/slide_1_5039_12703.html (Tang et al., 2011).**

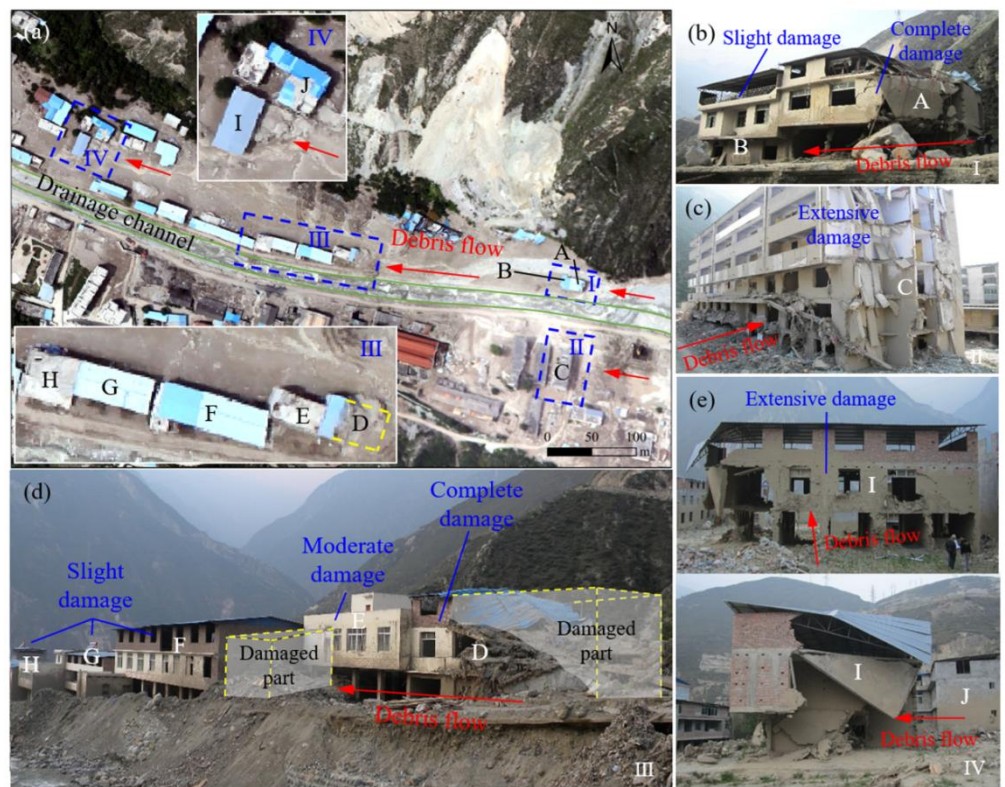

**Figure 8. Distribution of damaged-buildings in Qipan gully debris flow event. Orthophoto map is from the Basic Geographic Information Center of Sichuan Province, China, 21 July 2013.**

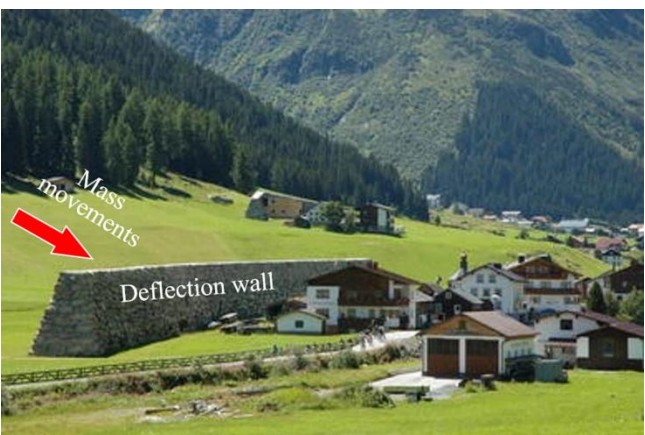

 **Figure 9. A deflection wall used to protect an entire building ensemble from the impact of medium magnitude events (Galtur Tschafein, Austria) (Holub et al., 2012).**

### 4.3 Effect of the distance to the surrounding buildings

As shown in Fig. 10, the variations in peak impact forces change with the surrounding buildings' distances in conjunction with the influences of azimuth angles. The scenario of the orientation of 90° and opening of 0.4 (*Or*90-*Op*0.4-*A*x-*D*x) is

355 taken as an example, and the background value (**Or**90-**Op**0.4-**A**null-**D**null) is still 3717 kN. According to the results of the sensitivity analysis, the most significant second-order effect comes from the interaction between the azimuth angle and distance, which can be divided into an amplification of the shielding effect or a reduction in the canalization effect, as shown in Fig. 10.

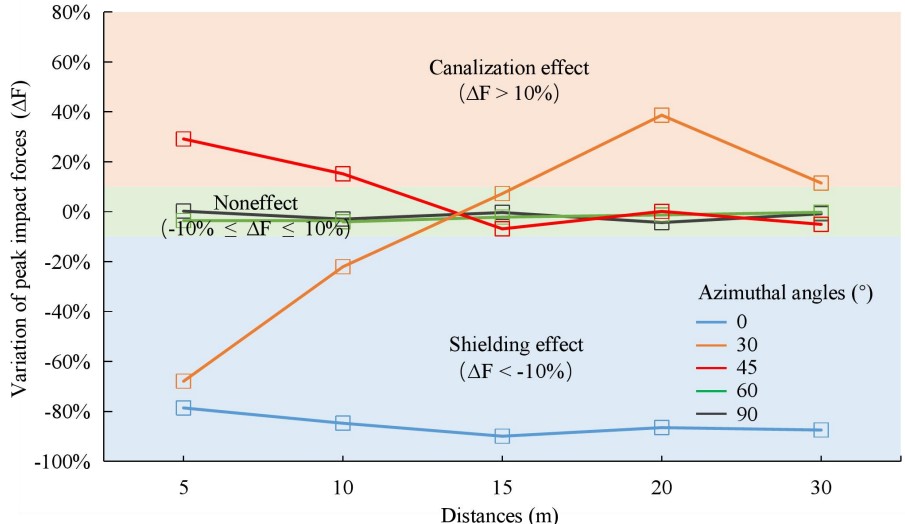

**Figure 10. Peak impact forces change with the surroundings' distances under influence of azimuth angles in the scenarios of orientation 90° and opening scale 0.4 (*Or*90-*Op*0.4-*A*x-*D*x).**

(1) Amplification of the shielding effect: The peak impact forces are found to be reduced gradually with increasing surrounding building distance in the case of an azimuth angle of 0° (**Or**90-**Op**0.4-**A**0-**D**x), with a 5 m distance yielding 796 kN and a 30 m distance yielding 467 kN, and the corresponding variations in peak impact forces are approximately -78.58%

and -87.44%. The shielding effects at an azimuth angle of 0° are amplified with increasing surrounding distance. This process occurs because there is a broader shielding area further downstream of the surrounding building when the debris flow path is separated by an obstacle at a fixed angle of spread, as shown in Fig. 11. Furthermore, there is likely a small-scale debris flow conflux zone close to the surrounding building, as shown in the red zone of Fig. 11. Buildings upstream, therefore, may suffer from a higher impact load in the same debris flow shielding area. As shown in region III of Fig. 8, the

lower a buildings' damage degree, the further away the completely damaged building D is in the back shielding area. Building E, next to building D, is exposed to moderate damage, and other shielding-protected buildings, such as buildings F, G and H, experience only slight damage from lateral abrasion and accumulation. However, the amplified shielding effect inevitably disappears further downstream due to the confluence of debris flow run-off. The range of the shielding area mainly depends on the debris flow properties, especially the friction coefficient and dynamic viscosity (Liang et al., 2021).

Further investigation of the effective shielding-protection area is needed.

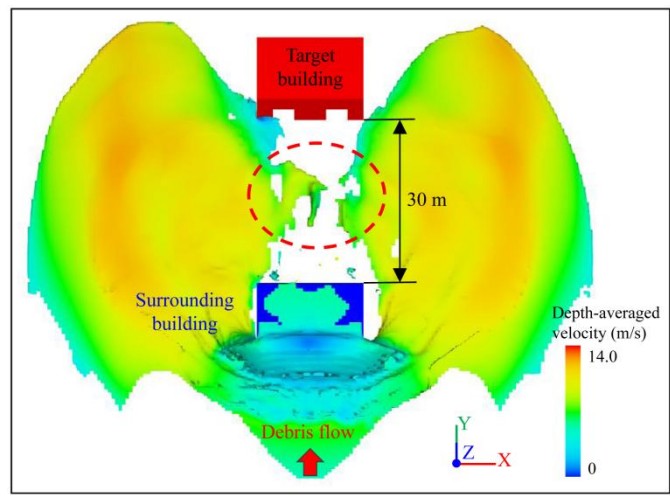

**Figure 11. There is the lower debris flow intensities further away surrounding building within a shielding area, in scenario of *Or*90-*Op*0.4-*A*0-*D*30 at simulation time of 8.0 s. The local debris flow conflux is delineated with red dotted line (The snapshot is rotating 30° counterclockwise about the X axis based on plan view, similarly hereinafter).**

(2) Reduction in the canalization effect: The canalization effect mainly occurs under a surrounding building azimuth angle of 45°, as mentioned above; however, this kind of effect could be lessened under the influence of the surrounding buildings' distances. As shown by the red line in Fig. 10, the maximum peak impact force under an azimuth angle 45°, with a value of 4800 kN, appears with a distance 5 m (*Or*90-*Op*0.4-*A*45-*D*5). Then, the peak impact forces decrease rapidly with greater distances, especially in the distance range of less than 15 m. The peak impact force at a distance of 15 m (*Or*90-*Op*0.4-*A*45-*D*15) is reduced to 3462 kN, which is very close to the background value (3717 kN). It is indicated that the canalization effect under an azimuth angle of 45° may have vanished completely at this point. As shown in the computational results at 8.0 s in Fig. 12, the increase in flow velocity in the narrowed flow paths due to building blockage decreases with increasing surrounding building distances. The increased-velocity debris flows, on the other hand, tend to flow away from the target building in the cases of greater surrounding building distances. Therefore, the variations in peak impact forces are close to 0 due to the lower flow velocity and fewer intruding materials in the scenarios of surrounding building distances beyond 15 m. In general, a greater distance results in a lower impact force.

A smaller surrounding building distance, however, does not necessarily indicate a larger impact force. With surrounding building distances of 5 m and 10 m, impact force shielding effects occur under the condition of an azimuth angle of 30° (*Or*90-*Op*0.4-*A*30-*D*5 and *Or*90-*Op*0.4-*A*30-*D*10), as shown in Fig. 13, and the corresponding variations in peak impact loads are -67.90% and -22.03%, respectively. The width of the narrowed flow path, which is determined by the factors of the azimuth angle and surrounding building distance, could account for this fact. The flow paths are so narrow, with widths of 1.6 m and 6.6 m, respectively, like bottlenecks, that not much flow passes through them, as shown in Fig. 13a and Fig. 13b. Thereafter, the canalization effect occurs at distances beyond 15 m and reduces and even disappears gradually with increasing surrounding building distance and the corresponding wider flow paths. It is found that the ratio of the width of the

narrowed flow path to the length of the target building has a significant effect on the increase in impact force. The largest peak impact force is most likely to be found under a ratio of approximately one, for example, the ratios of 0.8 in the case of *Or*90-*Op*0.4-*A*45-*D*5 (Fig. 12a) and 1.1 of *Or*90-*Op*0.4-*A*30-*D*20 (Fig. 13d).

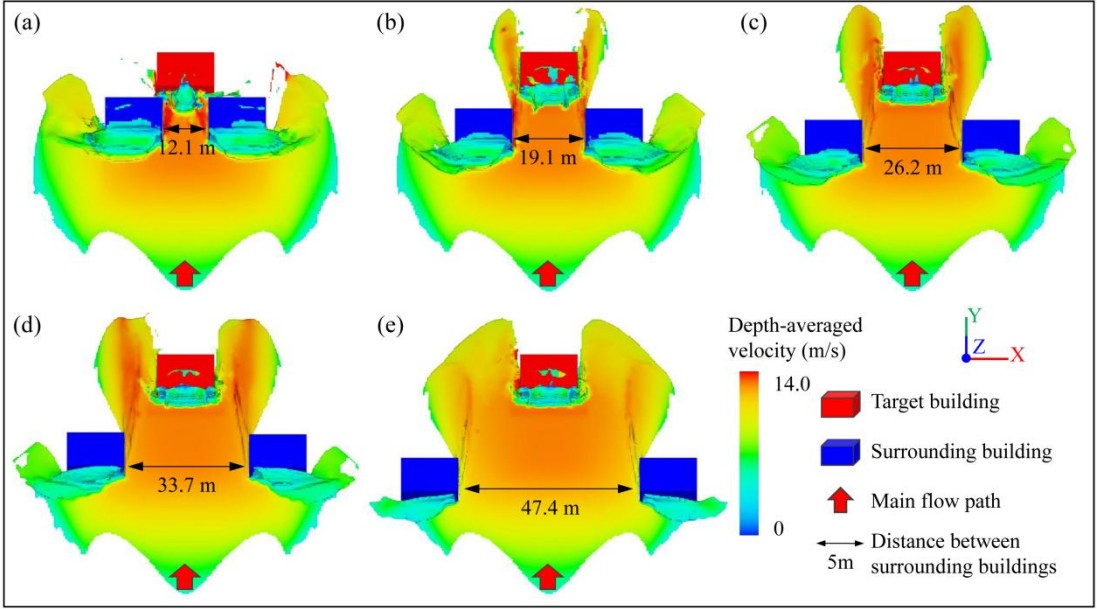

**Figure 12. Snapshots for debris flow intensities in scenarios of *Or*90-*Op*0.4-*A*45-*D*x at simulation time of 8.0 s. (a) is case of *Or*90-*Op*0.4-*A*45-*D*5; (b) is case of *Or*90-*Op*0.4-*A*45-*D*10; (c) is case of *Or*90-*Op*0.4-*A*45-*D*15; (d) is case of *Or*90-*Op*0.4-*A*45-*D*20; and (e) is case of *Or*90-*Op*0.4-*A*45-*D*30.**

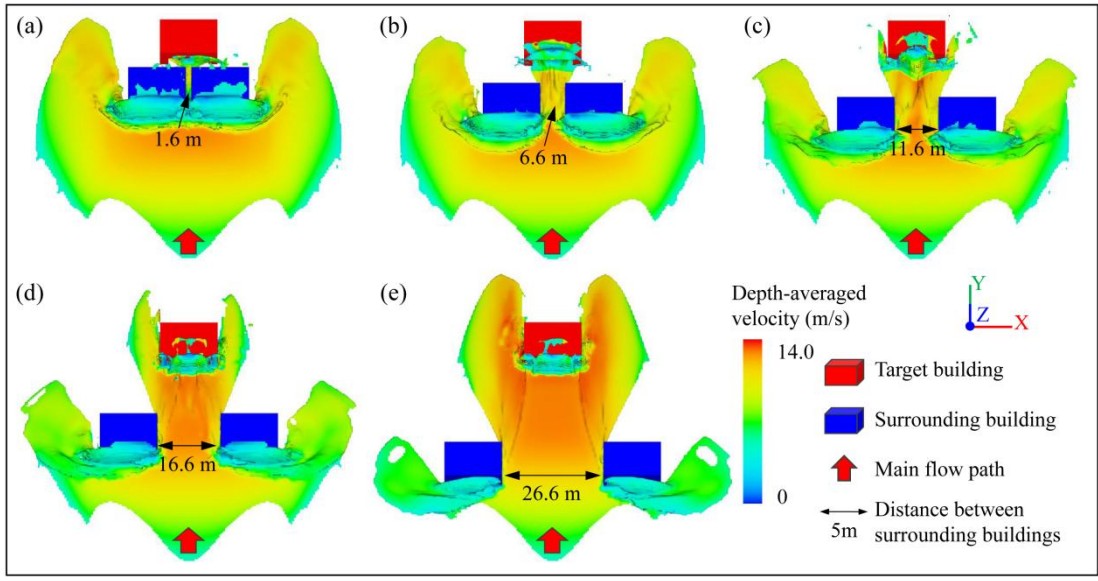

**Figure 13. Snapshots for debris flow intensities in the scenarios of *Or*90-*Op*0.4-*A*30-*D*x at simulation time of 8.0 s. (a) is case of *Or*90-*Op*0.4-*A*30-*D*5; (b) is case of *Or*90-*Op*0.4-*A*30-*D*10; (c) is case of *Or*90-*Op*0.4-*A*30-*D*15; (d) is case of *Or*90-*Op*0.4-*A*30-*D*20; and (e) is case of *Or*90-*Op*0.4-*A*30-*D*30.**

### 4.4 Effect of the orientation

#### 4.4.1 Single-factor analysis of orientation

As shown in Fig. 14, the peak impact forces of debris flows are ordered as follows: *Or*90 > *Or*60 > *Or*0 > *Or*45 ≈ *Or*30, and
the corresponding values are 3560, 2591, 2334, 1917 and 1912 kN, respectively, when only the target building's orientation
is considered (*Or*x-*Op*0-*A*null-*D*null). It is generally accepted that the impact load of a debris flow is a
comprehensive outcome from many characteristic parameters, including the debris flow density, velocity, impact contact
area and approaching angle (Liu et al., 2021). It is assumed that the debris flow density is constant in the
computational process of impact forces in the FLOW-3D model.

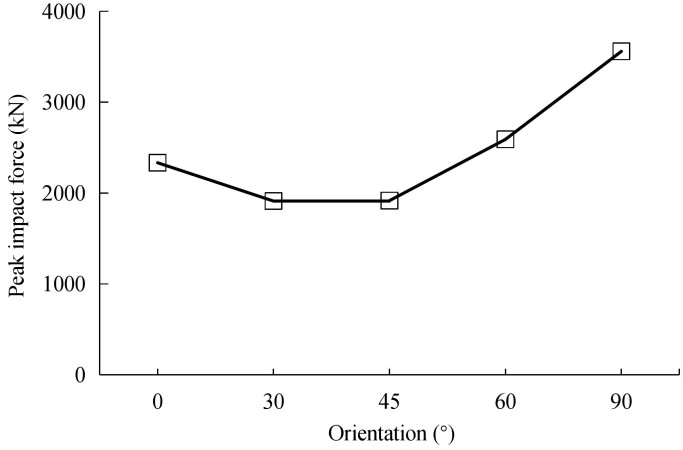

Figure 14. Peak impact forces change with target building's orientations in the scenarios of no openings and no surroundings (*Or*x-*Op*0-*A*null-*D*null).

The impact contact area, a product of the wall length and effective height, where the latter is the minimum value between the
wall height and flow depth, can be used to explain the debris flow impact responses in the case of orientations of 0° (*Or*0-*Op*0-*A*null-*D*null) and 90° (*Or*90-*Op*0-*A*null-*D*null). In the simulations shown in Fig. 15a and Fig. 15e, the single wall
element, wall B or wall A, is vertically impacted by debris flows, and the surging flows go beyond the wall height. The
length of the impacted wall elements, therefore, should contribute to the difference in the peak impact pressures. The mainly
impacted wall element in the orientation of 90° - Wall A of 15 m - is obviously longer than that of orientation 0°, wall B of
10 m. A larger contact area results in a greater impact force.

Walls A and B are simultaneously exposed to debris flows in the orientations of 30°, 45° and 60°, as shown in Fig. 15b-d.
In these cases, the debris flow impact loads, to a large extent, are dominated by the approaching angle, which is defined here
as the general, temporally independent angle of the wall element to the main flow path, with a range of 0° (parallel) to 90°
(vertical). Generally, there are higher flow velocities and lower surge flow depths in the cases of smaller approaching
angles, and vice versa. The highest flow depth occurs in the neighborhood of wall A, the longest wall of the target building,
in the scenario with an orientation 60° (*Or*60-*Op*0-*A*null-*D*null), as shown in Fig. 15d. In contrast, the lowest flow depth

appears near wall A, with an orientation of 30° (*Or*30-*Op*0-*A*null-*D*null), as shown in Fig. 15b. This could be the main reason why the target building with a 60° orientation is under stronger strain than that with a 30° orientation, and that with a 45° orientation is between them. However, even so, better migration performances, such as lower impact loads and larger shielding areas, are produced in these cases than in the case with an orientation of 90°. Similar to this idea, a splitting wedge,
with a triangular shape and two downslope-directed sides, was constructed at the process-oriented side of an exposed building to protect against snow avalanches in the Swiss Alps, as shown in Fig. 16. It was confirmed that splitting wedges, with this very distinctive shape, were considerably effective in maintaining its stability and offering a larger protected zone for the other neighboring buildings. It also provides a good model for building protection design in debris flow prone areas. The main criterion for the effective operation of such a structure is avoiding the highest flow depth - the maximum
approaching angle - appearing near the longest wall element.

Last but not least, flip-though impacts, caused by the backwater effect, which is a special phenomenon when debris flow hits a barrier wall, runs up, bounces backward, blocks and converges with the remaining debris (Takahashi, 2007; Song et al., 2021), contribute greatly to the peak impact forces of orientations 0° (*Or*0-*Op*0-*A*null-*D*null) and 90° (*Or*90-*Op*0-*A*null-*D*null), as shown in Fig. 17. The impact forces with an orientation of 0° reach their peak when the debris flow first collides
with the wall and runs up. The peak impact load with an orientation of 90° comes from the secondary waves overtaking the flow front, after the flow bounces off the wall, collides and converges with the flow approaching from behind (Iverson et al., 2010; Choi et al., 2018).

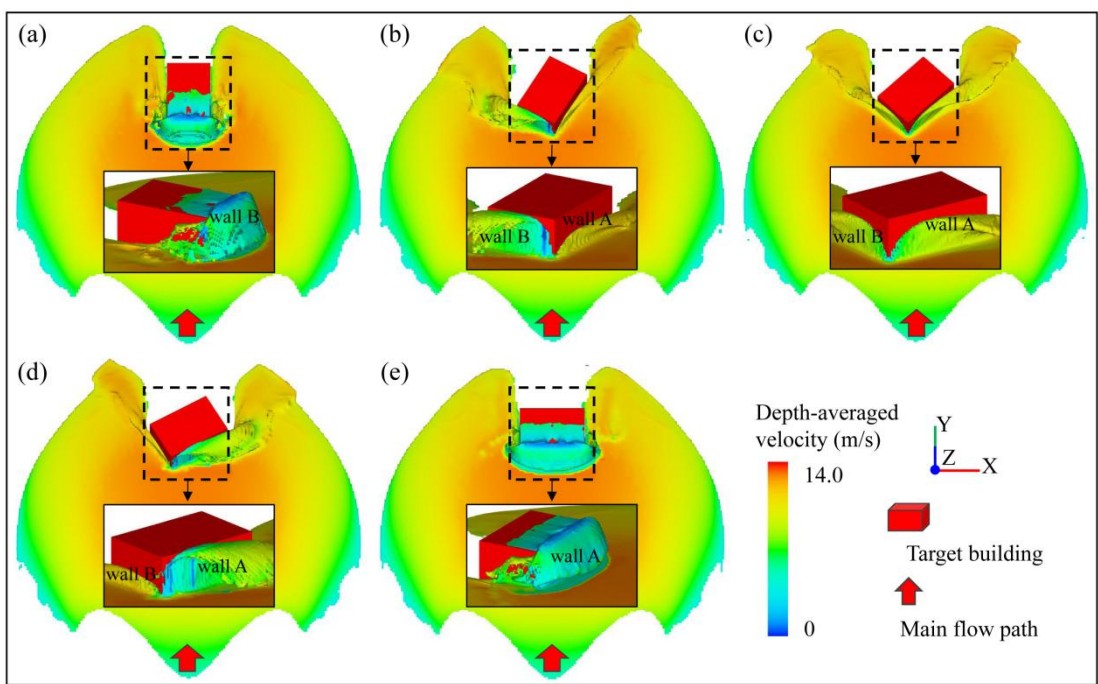

**Figure 15. Snapshots for debris flow intensities in scenarios of *Or*x-*Op*0-*A*null-*D*null at simulation time of 8.0 s. (a) is case of *Or*0-**
***Op*0-*A*null-*D*null; (b) is case of *Or*30-*Op*0-*A*null-*D*null; (c) is case of *Or*45-*Op*0-*A*null-*D*null; (d) is case of *Or*60-*Op*0-*A*null-*D*null;**
**and (e) is case of *Or*90-*Op*0-*A*null-*D*null.**

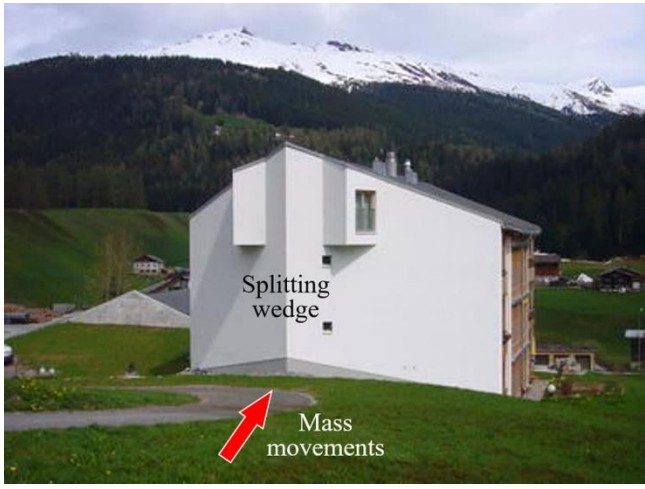

**Figure 16. Splitting wedge directly connected to the exposed object (Davos Frauenkirch, Switzerland) (Holub et al., 2012).**


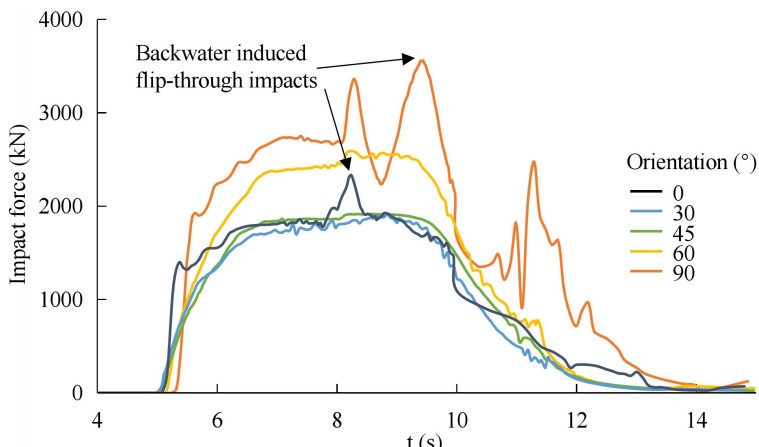

**Figure 17. Impact forces time history with changing orientations in the scenarios of *Or*x-*Op*0-*A*null-*D*null after simulation time of 4.0 s. The time of peak impact force are marked with dot.**

### 4.4.2 Interaction between orientation and surroundings

The single-factor analysis of orientation explained why the buildings with orientations of 30°, 45° and 60° have better migration effects. This knowledge has to be reconsidered, however, when the effect overlaps with the surrounding buildings' shielding effects. It is found that buildings with orientations of 30°, 45° and 60° are more likely to be damaged by debris flows within a shielding area. For instance, the maximum peak impact forces with surrounding buildings' distances of 5 m (*Or*60-*Op*0.4-*A*0-*D*5), 10 m (*Or*60-*Op*0.4-*A*0-*D*10) and 15 m (*Or*60-*Op*0.4-*A*0-*D*15) as shown in Fig. 18, are different from

those of the single-factor analysis of orientation. The protruding parts of the target building caused by the changing

orientations significantly contribute to the increase in impact forces (Hu et al., 2012; Zeng et al., 2015). A larger protruding portion of a building results in a greater probability of being exposed and corresponding larger impact forces. The largest protruding areas are exposed to debris flows at an orientation of 60°, as shown in Fig. 19d-1 and Fig. 19d-2). This can be confirmed by Fig. 8d. Although the main structure of building E is in the shielding area of building D, its protruding part is still completely destroyed by debris flows.

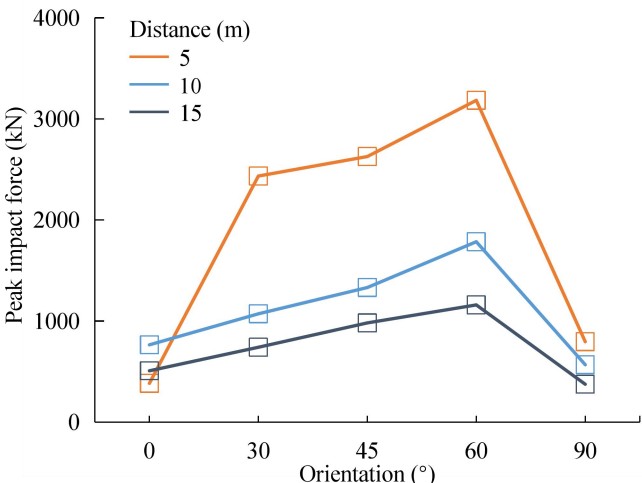

**Figure 18. Peak impact forces change with target building's orientations under the shielding effect. Red line shows the case of *Orx-Op*0.4-*A*0-*D*5; Blue line shows the case of *Orx-Op*0.4-*A*0-*D*10; Red line shows the case of *Orx-Op*0.4-*A*0-*D*15.**

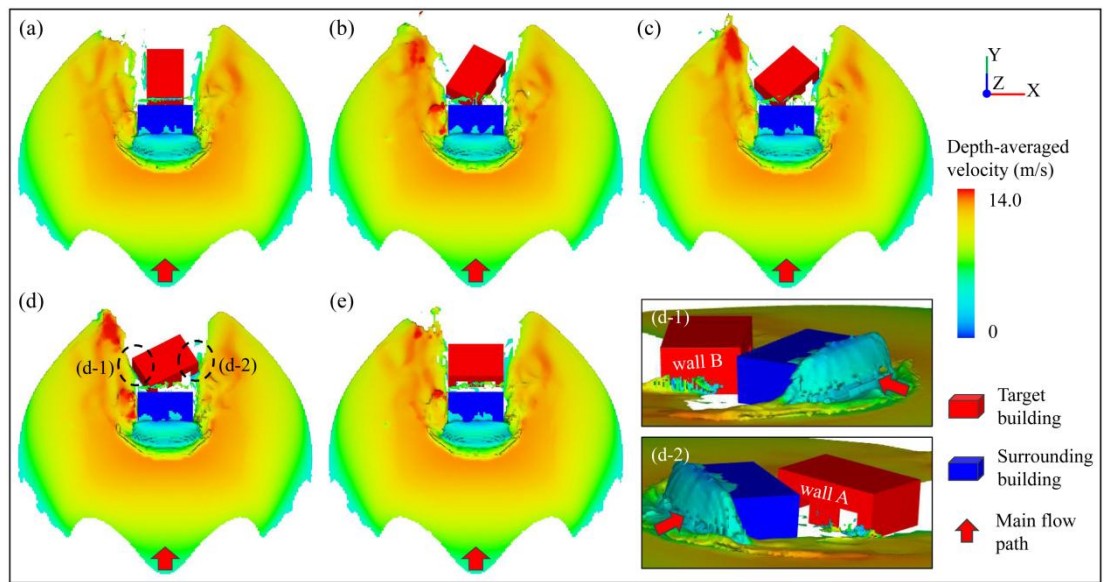

**Figure 19. Snapshots for debris flow intensities in scenarios of *Orx-Op*0.4-*A*0-*D*5 at simulation time of 8.0 s. (a) is case of *Or*0-*Op*0.4-*A*0-*D*5; (b) is case of *Or*30-*Op*0.4-*A*0-*D*5; (c) is case of *Or*45-*Op*0.4-*A*0-*D*5; (d) is case of *Or*60-*Op*0.4-*A*0-*D*5; (e) is case of *Or*90-*Op*0.4-*A*0-*D*5; (d-1) and (d-2) show that the protruding portions of target building are exposed to debris flow in details.**

**4.5 Effect of the opening scale**

According to the sensitivity analysis results, the opening scale is the least important factor for debris flow impacts, as it yields the minimum first-order effect index, 0.0050, and the minimum total effect index, 0.0137. From the results with an orientation 90° and no surrounding buildings (*Or*90-*Op*x-*A*null-*D*null), shown in Fig. 20, the peak impact forces of the target building change very slightly with increasing opening scale. There is a maximum peak impact force of 3945 kN in the case of openings with a scale of 0.8 (*Or*90-*Op*0.8-*A*null-*D*null), which is approximately 15.38% larger than the 3419 kN of

openings with a scale of 0.2 (*Or*90-*Op*0.2-*A*null-*D*null). Interestingly, the impact responses of the target building are different with different opening scales. Specifically, there are smaller impact forces in the cases of larger opening scales when only wall A of the target building is impacted, that is, in the early stage of debris flow impact from 5.3 s to 6.2 s, as shown in Fig. 21. In this stage, the maximum impact pressure of 865 kN of opening scale 0.8 is approximately half of the 1618 kN of opening scale 0.2; this is due to the difference in the effective impacted areas. From this perspective, the

mitigation performance of single wall elements with more openings is proven (Mazzorana et al., 2014; Gems et al., 2016). Thereafter, the impact load of the overall building increases rapidly following the abundant intrusion of materials through openings after 6.2 s, as shown in Fig. 21. Due to the greater accessibility and higher flow velocity, there is faster growth in impact pressure in the scenarios with the larger opening scales. Finally, after the two above-described impact stages are combined, there are only slight differences between multiple scales of openings in terms of peak impact forces. This

indicates that the mitigation function of openings for the whole building is very limited if the time for materials intrusion is sufficient.

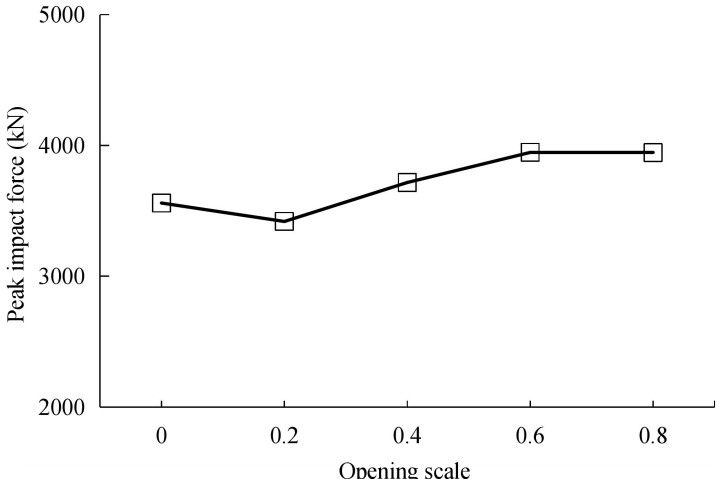

**Figure 20. Peak impact forces change with target building's opening scale in the scenarios of *Or*90-*Op*x-*A*null-*D*null.**

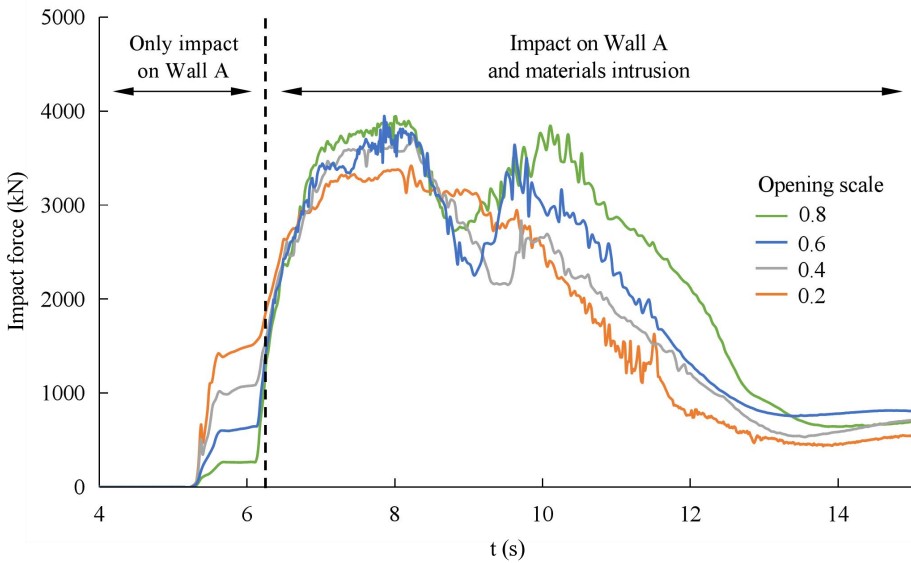


**Figure 21. Impact forces time history with changing opening scales in the scenarios of *Or*90-*Op*x-*A*null-*D*null after simulation time of 4.0 s. The time of peak impact force are marked with dot.**

## 5 Conclusions and outlook

The effects of representative built environment parameters on the debris flow impacts on a whole building were explored
through FLOW-3D simulations after validation with a published dam-break experiment. Four parameters influencing the
impact responses of the whole building induced by debris flows were considered in this study: the orientation and opening
scale of the target building and the azimuthal angle and distance of the surrounding buildings. The debris flow impact
performance was evaluated using the measurable indicator of the peak impact force acting on the overall building. It is
important to stress that the presented study and results were subject to a number of assumptions and limitations: (1) the type
of debris flow was limited as mudflow or viscous debris flow, the solid was assumed to be mixed with the fluid phase well,
and the sediment deposition was not considered; (2) the deposition fan was simplified for the modelling, for example the
drainage channel and bed scour had been ignored; (3) the inflow condition was different with the realistic debris flow
hydrograph, the discharge was fixed as 500 m$^3$ s$^{-1}$ and duration was limited to 5s, and the peak impact force was treated as
the maximum value within the computation time of 15 s. Finally, the main outcomes of the study may be outlined as follows:
(1) The GSA based on metamodels with 160 cases reveals that the ranking of the importance of the built environment
parameters on debris flow impacts from the results of total effect indices is as follows: azimuth angle (*A*) > distance (*D*) >
orientation (*Or*) > opening scale (*Op*). The azimuth angle of the surrounding buildings alone contributes to 63.03% of the
overall variance of the debris flow peak impact load. The properties of the surrounding buildings, including the azimuth
angle and distance, are found to have a more significant influence on the peak impact forces.

(2) The azimuth angle has a shielding or canalization effect on debris flow impacts. The shielding effect, a form of reducing impact pressures, mainly appears in the scenarios with a surrounding building azimuth angle of 0°. The canalization effect, caused by narrowing and redirecting of the flow path, is a form of increasing impact forces and occurs at an azimuth angle of 45°. A deflection wall for building protection is recommended, as this provides a shielding effect. The interaction between the azimuth angle and distance can be divided into the amplification of the shielding effect and the reduction in the

canalization effect. The former is where buildings are less impacted with a limited increase in distance within a shielding area. Further investigation on the effective area of shielding protection is needed. The latter is where the peak impact force induced by the canalization effect decreases rapidly with a greater distance. The ratio of the width of the narrowed flow path to the length of the target building has a significant effect on the variation in the impact forces, and the maximum peak impact pressure appears at a ratio of approximately one.

(3) These parameters involving the building's impact response, including the impact contact area, approaching angle and flip-though impact, contribute to the debris flow impact forces when only the orientation factor is considered. A splitting wedge is recommended for an effective design mitigating the threat of debris flow, and the main criterion is avoiding the highest flow depth - the maximum approaching angle - appearing near the longest wall element. The buildings with orientations of 30°, 45° and 60° are more likely to be impacted by debris flows in a shielding area due to the exposed

protruding parts produced by the building's rotations. As far as openings are concerned, although the mitigation performance of this single wall element has been proven, a limited effect on the whole building is observed when there is enough time for material intrusion.

It is obvious that the quantitative descriptions about the interactions between the built environment and impact forces can be useful to the built environment improvement and local adaptation measures for the impact force reduction, which are

assumed as the low-cost and efficient approach for mitigating the building's structural damages. And more significantly, the present paper has extended the knowledge about the influence factors on debris flow intensity. It is demonstrated that some artificial building factors can not be ignored, except for the natural environments, in deciding the spatial pattern of the process intensity. The further research about their relative importance with the 3-D numerical simulation and sensitivity analysis can promote the relative intensity evaluation of the building, especially in terms of the indicator selection and

weighting, which may open a future topic of the debris flow hazard assessment. For the building vulnerability assessment, the indicators can be mainly divided into two kinds: the exterior process intensity and interior building resistance. The process intensity, for example the flow depth, velocity, impact force or the other proxy, was assumed absolutely necessary, either in the curve based approach or the indicator based approach (Martinez-Carvajal et al., 2018). From the current literature, however, there are some confusions in selecting the surroundings factors and process intensity indicator. To be

specific, some surroundings factors or also called protection factors, including the Surrounding buildings, Building row, Wall around building, Natural barriers and so on, were still selected when the debris flow intensity had been indirectly considered (Dall'Osso et al., 2009; Dall'Osso et al., 2016; Papathoma-Köhle et al., 2019). These indicators should be independent each other theoretically. From the views of the present paper, the functions of all over the surroundings factors

are the influences on the process intensity around building. Therefore, the process intensity should be exclusive with the

surroundings factors. The building features factors are mainly considered to be acted on the building resistance, including the

Material, Structure, Number of stories, Foundation strength and so on. However, It is not hard to find that some building

indicators, for example the Orientation, Shape and Openings, can rebuild the process intensity. As a result, the effect of the

representative building features indicators on the building vulnerability needs an in-depth discussion in future. The last but

not least, a more universal, robust index may be developed using the numerical simulation approach, which can improve the

locality limits resulting from the empirical data, to some extent.

*Author contribution.* XH and ZZ contributed to the original idea and study design. XH, ZZ and GX participated in field survey. XH and ZZ conducted the simulation and analyse. XH wrote the original manuscript, and ZZ and GX provided comments and revised the manuscript. All the co-authors contributed to scientific interpretations of the results.

*Competing interests.* The authors declare that they have no conflict of interest.

*Financial support*. This research has been supported by the National Natural Science Foundation of China (grant No. 41907396), the Science and Technology Research Program of Chongqing Municipal Education Commission (grant No. KJQN201900535), the Chongqing Normal University Founding Program (grant No. 21XWB007), the Chongqing Normal University Postgraduate Research and Innovation Project (grant No. YKC21047) and the Scientific Research Project of Department of Natural Resources of Sichuan Province (grant No. KJ-2021-14).

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
