# Peer review of "Sensitivity analysis of a built environment exposed to the synthetic monophasic viscous debris flow impacts with 3-D numerical simulations"

_Natural Hazards and Earth System Sciences, 2022_

## Referee Comment (RC1)

**Review of manuscript NHESS-2022-173**

**Sensitivity analysis of a built environment exposed to debris flow impacts with 3-D numerical simulations.**

**Overview**

In the manuscript, the authors analysed how the surrounding buildings modify the maximum debris-flow impact force on a target building. The purpose was to identify how some geometrical features of the buildings affect the impact force. The authors used the three-dimensional commercial code FLOW-3D to investigate numerically them. Firstly, they validated the code with a flume experiment, and then 160 simulations were performed and used for metamodel modelling. This approach allowed defining a ranking among the geometrical feature involved. Finally, additional simulation was performed and analysed to understand if and how the surrounding building modifies the debris-flow impact force.

**Observations and critical points**

In the following, I list my main observations and critical points about the manuscript.

1.  Debris flow is a word commonly used to describe the flow of a mixture composed of water and a high concentration of sediment. Depending on the type and quantity of finer sediment grains, debris flow can be dived into two main classes: stony debris flow where the percentage of cohesive material (usually clay and the finer classes of silt) is negligible and mudflow where the percentage of cohesive sediment is important. This division is necessary since the rheological properties of the two are quite different: in the mudflow, the solid particles are essentially suspended inside fluid and the division between solid and fluid are difficult. This means that a monophase approach can be used and the mixture presents a non-Newtonian behaviour where yield stress is present (e.g., Bingham fluid). On the other hand, in the stony debris flow the two phases are easily identified and divided. This implies that a two-phase approach is necessary where the fluid behaviour is usually Newtonian, while the solid phase presents a collisional regime. See e.g., Iverson 1997, Takahashi 2007, Armanini 2013. The authors have to clarify which type of debris flow are dealing with: it seems, from the validation test that a stony debris flow is the target, however, in all the other sections it seems that a mudflow is analysed.

2.  The authors use the FLOW-3D code to simulate debris flow.  Which are the equations used? These are crucial when you describe the parameters used. Without the equations, the parameters described by the author can be not present in the model. Moreover, looking at other papers dealing with FLOW-3D, also other parameters are needed: these are not listed in the manuscript.

3.  A peculiarity of stony debris flow is the rapid formation of large scour and deposition. Deposition rapidly occurs when the mixture flow decreases the velocity, while scour usually happens when the flow is accelerated. Both decreasing and increasing velocities are present when the flow impacts the building. From the literature, I found that the model FLOW-3D can describe scouring and deposition for river and coastal morphodynamics, so when the morphological variation presents a longer time scale than the hydrodynamic one. In debris flow, the variation is of the same time scale. How did the authors consider this bed variation in the FLOW-3D modelling?

4.  How are the impact forces evaluated? The authors write "the General Moving Objects (GMO) model of FLOW-3D was applied to obtain the overall impact forces on the building, in which a rigid body motion was introduced for the fluid-rigid interaction behaviour (Postacchin, 2019; Isobe, 2021)". Is it correct that the object where the forces are evaluated must be in motion? How is it possible to use this when dealing with a fixed and non-deformable target building as the one described in the manuscript?

Moreover, the citations proposed are not relevant: Postacchini 2019 deals with experimental apparatus where a movable reference system is used (they move the building in a static pool of water), while Isobe 2021 deals with movable and deformable steel frame buildings but with another kind of models, not the FLOW-3D.

5. In the validation section, the authors reproduce one laboratory experiment. The particular stony debris flow experiments can be reproduced well also with a monophase approach since the bed is rigid and all the material remains quite well mixed during all the experiment (only some separation between solid and fluid phase is visible in figure 2(c)). However, I think that it is not correct to say "FLOW-3D reproduces the debris flow impact process in the flume test very well" basing the statement mainly on the peak impact pressure. It is important also the time history of the pressure: arrival time of the flow, the timing of the peak, duration of the peak, etc. Moreover, it is missing some parameters used in the model (e.g., the roughness) and it is not clear the dimension of the cell: is it composed of cubes of 0.02 m side? If yes, since the first load cell position is 0.015 m from the bottom of the flume, how do the authors evaluate the pressure at that height that is neither on the centre nor the border of the cube? Additionally, on line 105 the authors highlight that the data is averaged over 10 points (it means cells?) how is it possible to do this in the flume experiment? Is it horizontally averaged? Finally, for better validation, I suggest using the calibrated parameter to reproduce a second flume experiment and discuss it.

6. In the numerical modelling, the authors used a fixed discharge of 500 m³/s for a very short time (10 s). If the peak of discharge could be of some interest for very large debris flow, however, the duration and the constant value are not realistic and leads to unrealistic values of impact force. A more realistic debris flow inflow can be a triangular one where the overall duration is about 15 minutes with a peak discharge that occurs after 5 minutes (some examples of real and simplified hydrographs with can be found in Berger & al. 2011, Marchi & al. 2021). This modification in the inflow is essential for a truthful analysis of forces since, one of the main features of a debris flow just described previously, is the great deposition that occurs when the flow is slowed down. The direct consequence of the deposition is the time increase of the pressure due to this saturated terrain at rest.

7. Some perplexity will arise also by looking at some of the parameters used: roughness and viscosity. For the surface roughness, the authors used 0.05 m which represents "the equivalent grain roughness (or absolute height in meters)". This means that on all the surfaces of the computational domain (that also includes the buildings) the roughness is generated by grains of 5 centimetres. This kind of roughness can be representative of a natural environment (e.g., riverbed, grassland, wood) but in an urban environment, where usually the surfaces are paved or made of gravel, is too big. For the viscosity, the authors used 1 Pas. This value is at least one order of magnitude higher compared to the ones described by Iverson 1997 (the fluid viscosity ranges from 0.001 Pas to 0.1 Pas) or also the ones measured by Song & al. 2021 (laboratory experiment with fluid viscosity ranging from 0.001 up to 0.1). Also, the authors use a value of 0.1 Pas to validate the FLOW-3D model (based on one of the experiments of Song & al. 2021). Why this choice? If you validate the model with 0.1 Pas, also the other simulations should be performed with similar viscosity.

8. The target building has walls with a thickness of 0.35 m (line 156), while the cell (cube?) has a 0.25 m side (line 158). How is possible to simulate a wall that has a dimension that is not a multiple of a cell? Why not use a wall thickness equal to the cell side?

9. The force is a vector, so it has an orientation. In the paper, I suppose, the authors report only its module. This aspect gives rise to two main questions. The first one is how the impact force is evaluated: is it evaluated also considering the tangential stresses on the walls? The second question is about where the force is evaluated: it is all over the surfaces of the building (inside and outside walls)? If the answer is yes, is it simply a sum of the force exerted by the mixture over all the walls? In this specific case, if there is flow inside the building, is the force on one wall the net force evaluated between inside and outside or is it the sum of the two? Moreover, is it considered also the roof?

10. When the azimuth angle A decreases and approaches 0, it has to be specified that the two surrounding buildings become a single building. Regarding this aspect, is the metamodel able to consider this? Otherwise, the authors have to be neglected, from the metamodel simulations, all the cases when the surrounding buildings are merged.
11. Regarding the metamodel simulation, what are the ranges of variation of the four input variables?
12. I think that the duration of the simulation, which, from figures 17 and 21 it is set to 10 s (as the discharge duration), is too short since it for some tests the maximum value of the impact force is registered at the end of the simulation when a positive trend is also visible. I suggest increasing the simulation duration until the mixture is fully stopped or is flowed away from the target building.

**Bibliography**

[1] A. Armanini, Granular flows driven by gravity, Journal of Hydraulic Research. 51 (2013) 111–120

[2] C. Berger, B.W. McArdell, F. Schlunegger, Direct measurement of channel erosion by debris flows, Illgraben, Switzerland, Journal of Geophysical Research: Earth Surface. 116 (2011)

[3] R.M. Iverson, The physics of debris flows, Reviews of Geophysics. 35 (1997) 245–296

[4] L. Marchi, F. Cazorzi, M. Arattano, S. Cucchiaro, M. Cavalli, S. Crema, Debris flows recorded in the Moscardo catchment (Italian Alps) between 1990 and 2019, Natural Hazards and Earth System Sciences. 21 (2021) 87–97

[5] D. Song, X. Chen, G.G.D. Zhou, X. Lu, G. Cheng, Q. Chen, Impact dynamics of debris flow against rigid obstacle in laboratory experiments, Engineering Geology. 291 (2021) 106211

[6] T. Takahashi, Debris flow: mechanics, prediction and countermeasures, Taylor & Francis, 2007

---

## Referee Comment (RC3)

**Some comments on the Authors response to the Reviewer's questions for manuscript nhess-2022-173**

I appreciate the effort done by the authors to answer the question that I propose. However, maybe due to a not well-posed question, some questions are still unclear to me.

The main one concern the validation. In the laboratory experiment used for the validation, a well-mixed stony granular debris flow is reproduced. One of the main characteristics of this debris flow is that the energy dissipation is due to the collision between the particles and not by the viscosity of the fluid (e.g. Iverson 1997, Takahashi 2007, Armanini 2013). However, in the authors' response to my comment #1, it is highlighted that "From the characteristics of RNG k-ε model [that is used in all the manuscript], the type of debris flow involved in this study was determined as mudflow or viscous debris flow, in which a single-phase non-Newtonian fluid was assumed and solid particles were treated as suspension and mixed with the fluid phase well". This statement is completely in contrast with the used laboratory experiment used and consequently, all the section devoted to the validation of the model is meaningless since the author used a model that could not represent correctly the physical processes involved.

Another point of the validation part regards why the authors do not show the time history of the impact force. Since one of the characteristics of a debris flow impact process is its dynamic changes in time as the experiments of Song et al. 2021 show (the time history is quite complex and is not only represented by a single value!), the "simple" peak value is not sufficient for validating the model used. For this reason, I think that the authors' response "It is demonstrated that the RNG and GMO coupled model in FLOW-3D are able to describe the peak impact force and fluid surface effectively" is not fully trustable.

A third comment regards the author's answer to comment #6 in combination with #12. I know well that long simulations use a high quantity of memory and take long computational times, so for this reason it could be, in some cases, acceptable to use high fixed discharge for a short time. However, I think that the 10-second duration used by the authors is not fully appropriate at least for some of the simulations used. For example, it is clear from Figure 17 that for the simulation with 45° of orientation (Or45) the peak impact force is the last value of the plot (i.e. at 10 second, so at the end of the simulation) but the force has a trend is still increasing! Also for the cases of 60° and 30° (i.e Or60 and Or30), the trend of the force is still increasing and at the end of the simulation (i.e. the end of the plot), the values are very close to the peak values. This means that, at least, in these three simulations (but I think that the same problem arises also in lots of other simulations done by the authors as the one shown in Figure 21) the authors have to increase the time of the simulation until a significant (a few seconds?) decreasing, or at least constant, value of the impact force is visible.

The last comment regards the author's response to comment #9 regarding the force. For me, it remains unclear the meaning of the number that represents the force. Moreover, since the target building is a complex geometry, where these "numbers" are applied? It is quite different if this "number" is applied only to a single surface (e.g. only on the wall with the opening) or it is applied with different values on different surfaces (e.g. on the wall with the opening plus the roof) because the possible consequences are completely different! Here, I speak about "number" since, as in comment #9, I underline that the force is a vector, so a simple "number" does not represent the force: it is still missing the direction of this force and the point where the force is applied!

---

## Author Comment (AC1)

**RESPONSE TO REVIEWER COMMENTS:**

**Authors' General comment:** Thank you for the Reviewer's constructive comments concerning our manuscript entitled "Sensitivity analysis of a built environment exposed to debris flow impacts with 3-D numerical simulations" (ID: nhess-2022-173). We believe the revised version of the manuscript, which addresses the Reviewer's comments, is now more consistent with the current literature and clarifies the important points raised by the Reviewer. Revised portions are marked in red in the manuscript and response letter, and the manuscript is re-submitted in clean format to the Journal. Please also find below my response to Reviewer comments.

**REVIEW COMMENTS:**

**1. Debris flow is a word commonly used to describe the flow of a mixture composed of water and a high concentration of sediment. Depending on the type and quantity of finer sediment grains, debris flow can be dived into two main classes: stony debris flow where the percentage of cohesive material (usually clay and the finer classes of silt) is negligible and mudflow where the percentage of cohesive sediment is important. This division is necessary since the rheological properties of the two are quite different: in the mudflow, the solid particles are essentially suspended inside fluid and the division between solid and fluid are difficult. This means that a monophase approach can be used and the mixture presents a non-Newtonian behaviour where yield stress is present (e.g., Bingham fluid). On the other hand, in the stony debris flow the two phases are easily identified and divided. This implies that a two-phase approach is necessary where the fluid behaviour is usually Newtonian, while the solid phase presents a collisional regime. See e.g., Iverson 1997, Takahashi 2007, Armanini 2013. The authors have to clarify which type of debris flow are dealing with: it seems, from the validation test that a stony debris flow is the target, however, in all the other sections it seems that a mudflow is analysed.**

**AUTHORS RESPONSE:** We greatly appreciate for Reviewer's good comments! As Reviewer suggested, the division about which type of debris flow involved is indeed very necessary. We added the statements about debris flow type in Line 114-116 of the revised manuscript.

**Line 114-116:**

From the characteristics of RNG k-ε model, the type of debris flow involved in this study was determined as mudflow or viscous debris flow, in which a single-phase non-Newtonian fluid was assumed and solid particles were treated as suspension and mixed with the fluid phase well.

**2. The authors use the FLOW-3D code to simulate debris flow. Which are the equations used? These are crucial when you describe the parameters used. Without the equations, the parameters described by the author can be not present in the model. Moreover, looking at other papers dealing with FLOW-3D, also other parameters are needed: these**

**are not listed in the manuscript.**

**AUTHORS RESPONSE:** Thanks for Reviewer's good suggestion! In this study, the RNG k-ε model of FLOW-3D was applied to simulate the transportation process of debris flow. As Reviewer suggested, the statements about RNG model descriptions, equations and main parameters was added in Line 96-101 and Line 105-112 of the revised manuscript.

**Line 96-101:**

In this study, the renormalized group (RNG) model-based k-ε turbulence model and the general moving objects (GMO) model are applied to build fluid-solid coupled model of the debris flow impact. The RNG k-ε model is a modification of the standard k-ε model, which takes the turbulent vortex into account and provides an analytic formula for Prandtl number, as well as an analytic formula for low Reynolds number flow viscosity (Franco et al., 2021). These features make the RNG model more reliable and accurate for a broader flow than the standard k-ε model (Yin et al. 2015).

**Line 105-112:**

The two transport equations of RNG k-ε model in FLOW-3D are as following:

$$\frac{\partial k_T}{\partial t} + \frac{1}{V_F}\left\{uA_x\frac{\partial k_T}{\partial x} + vA_y\frac{\partial k_T}{\partial y} + wA_z\frac{\partial k_T}{\partial z}\right\} = P_T + G_T + Diff_{k_T} - \varepsilon_T \tag{1}$$

$$\frac{\partial \varepsilon_T}{\partial t} + \frac{1}{V_F}\left\{uA_x\frac{\partial \varepsilon_T}{\partial x} + vA_yR\frac{\partial \varepsilon_T}{\partial y} + wA_z\frac{\partial \varepsilon_T}{\partial z}\right\} = \frac{CDIS1\cdot\varepsilon_T}{k_T}(P_T + CDIS3\cdot G_T) + Diff_\varepsilon - CDIS2\frac{\varepsilon_T^2}{k_T} \tag{2}$$

where $k_T$ is the turbulent kinetic energy, $V_F$ is the fractional volume open to flow, $A_x$, $A_y$ and $A_z$ are the fractional area open to flow in the x, y and z directions, respectively. $P_T$ is the turbulent kinetic energy production term, $G_T$ is the buoyancy production term, *Diff* is the diffusion term, and $\varepsilon_T$ is the turbulence dissipation term. In the RNG model of FLOW-3D, *CDIS*1 and *CDIS*3 are dimensionless user-adjustable parameters that have defaults of 1.42 and 0.2, respectively, and *CDIS*2 is determined from $k_T$ and $P_T$ (Flow Science, Inc., 2014).

**3. A peculiarity of stony debris flow is the rapid formation of large scour and deposition. Deposition rapidly occurs when the mixture flow decreases the velocity, while scour usually happens when the flow is accelerated. Both decreasing and increasing velocities are present when the flow impacts the building. From the literature, I found that the model FLOW-3D can describe scouring and deposition for river and coastal morphodynamics, so when the morphological variation presents a longer time scale than the hydrodynamic one. In debris flow, the variation is of the same time scale. How did the authors consider this bed variation in the FLOW-3D modelling?**

**AUTHORS RESPONSE:** We thank Reviewer very much for the comments! As Reviewer suggested, the formations of large scour and deposition indeed have the significant effects on the impact forces of debris flow. As mentioned in Question 1, the type of debris flow involved in this study was mudflow or viscous debris flow, in which the particles were treated as suspension and mixed with the fluid phase well. The division between solid and fluid was assumed to be difficult, therefore, the granular deposition was not considered in this study (as

added in Line 116-117 of the revised manuscript). This deposition fan was treated as a rigid bed model, therefore, the bed material scour would not happen in this study. To sum up, the bed variation induced by sediment transport processes, including sediment deposition and bed scour, was not considered (as added in Line 176-178 of the revised manuscript).

In addition, as Reviewer and literature mentioned, the scouring and deposition for river and coastal morphodynamics can be indeed calculated using the Sediment Scour model, another physics model implemented in FLOW-3D code. They are done by considering two states in which sediment can exist: suspended and packed sediment. Suspended sediment is typically of low concentration and advects with fluid flow. And only a thin surface layer of grains of the packed sediment (in the thickness of a few grain diameters) can move in the form of bed-load transport (Flow Science, Inc., 2014). In order to maintain the computation stability, moreover, the sediment diameter should not exceed 10% of cell size. If a smaller grid is usually applied to describe the building geometry precisely, the maximum value of sediment diameter will be tiny accordingly. Obviously, the above-mentioned requirements limit the availability in realistic debris flow stimulation, to a certain extent.

**4. How are the impact forces evaluated? The authors write "the General Moving Objects (GMO) model of FLOW-3D was applied to obtain the overall impact forces on the building, in which a rigid body motion was introduced for the fluid-rigid interaction behaviour (Postacchin, 2019; Isobe, 2021)". Is it correct that the object where the forces are evaluated must be in motion? How is it possible to use this when dealing with a fixed and non-deformable target building as the one described in the manuscript? Moreover, the citations proposed are not relevant: Postacchini 2019 deals with experimental apparatus where a movable reference system is used (they move the building in a static pool of water), while Isobe 2021 deals with movable and deformable steel frame buildings but with another kind of models, not the FLOW-3D.**

**AUTHORS RESPONSE:** Thank you very much! We apologize for Reviewer's confusion induced by our incorrect statements and references. The statements about GMO model description, building setting and impact force calculation approach were added in Line 119-122 and Line 123-125 of the revised manuscript. In addition, the former irrelevant citations have also been deleted.

**Line 119-122:**

The GMO model can simulate the rigid body motion, which is either user prescribed or dynamically coupled with fluid flow. In this study, the target building and surrounding buildings were all prescribed to be the fixed and non-deformable rigid models. The hydraulic force and torque due to normal pressure and shear stress can be calculated at each time step.

**Line 123-125:**

The overall impact force of target building was gained from the combined fluid force considering the force direction in the GMO model, which was calculated from the normal pressures and shear forces in x, y and z directions.

**5. In the validation section, the authors reproduce one laboratory experiment. The particular stony debris flow experiments can be reproduced well also with a monophase approach since the bed is rigid and all the material remains quite well mixed during all the experiment (only some separation between solid and fluid phase is visible in figure 2(c)). However, I think that it is not correct to say "FLOW-3D reproduces the debris flow impact process in the flume test very well" basing the statement mainly on the peak impact pressure. It is important also the time history of the pressure: arrival time of the flow, the timing of the peak, duration of the peak, etc. Moreover, it is missing some parameters used in the model (e.g., the roughness) and it is not clear the dimension of the cell: is it composed of cubes of 0.02 m side? If yes, since the first load cell position is 0.015 m from the bottom of the flume, how do the authors evaluate the pressure at that height that is neither on the centre nor the border of the cube? Additionally, on line 105 the authors highlight that the data is averaged over 10 points (it means cells?) how is it possible to do this in the flume experiment? Is it horizontally averaged? Finally, for better validation, I suggest using the calibrated parameter to reproduce a second flume experiment and discuss it.**

(1) Response to comment: "it is not correct to say "FLOW-3D reproduces the debris flow impact process in the flume test very well" basing the statement mainly on the peak impact pressure.""

**AUTHORS RESPONSE:** We thank Reviewer for the very good advice! It is really true as Reviewer suggested that there are many indicators to describe the fluid impact process, except for the peak impact force. Therefore, the former statement that "FLOW-3D reproduces the debris flow impact process in the flume test very well" is indeed very biased. The revised statement is that "It is demonstrated that the RNG and GMO coupled model in FLOW-3D are able to describe the peak impact force and fluid surface effectively." (as added in Line 171-172 of the revised manuscript)

(2) Response to comment: "it is missing some parameters used in the model (e.g., the roughness) and it is not clear the dimension of the cell: is it composed of cubes of 0.02 m side?"

**AUTHORS RESPONSE:** Thanks for Reviewer's comments! The statements about the surface roughness parameter of the flume bed were added in Line 156-157 of the revised manuscript. And the supplementary statement about the dimension of the computation cell was added in Line 153-154.

**Line 156-157:**

In the physical experiment, the flume bed was roughened using 0.6 mm spherical glass beads, its surface roughness parameter was set as 0.0006 m accordingly in the validation model.

**Line 153-154:**

the targeted analysis domain was discretized into a grid with a cell size of 0.02 m, which was equal to a cube of 0.02 m side in 3-D model.

(3) Response to comment: "how do the authors evaluate the pressure at that height that is

neither on the centre nor the border of the cube?"

**AUTHORS RESPONSE:** We apologize for missing to describe the data collection approach of the load cells in FLOW-3D. As Reviewer mentioned, the load cell 1# was indeed set to be 0.015 m from the bottom of the flume, nearly the border of computation grid, however, this didn't affect its measured impact force. This is because that the load cell was treated as a history probe in FLOW-3D. History probes are point measurement tools and can be thought of as thermocouples or pressure transducers, which can allow to access specific information at a particular location (Flow Science, Inc., 2014) (as added in Line 167-169 of the revised manuscript).

(4) Response to comment: "on line 105 the authors highlight that the data is averaged over 10 points (it means cells?) how is it possible to do this in the flume experiment? Is it horizontally averaged?"

**AUTHORS RESPONSE:** We are very sorry for Reviewer's misunderstanding due to our inaccurate expressions. The impact force data was not averaged in the spatial dimension, but in the timeline. The raw data was collected at interval of 0.001 s from the numerical code. To reduce the uncertainty, a simple data noise reduction approach, that the peak impact forces were obtained from the average values over 10 points in the timeline (0.01 s), was executed (Song et al., 2021) (as added in Line 125-127 of the revised manuscript).

(5) Response to comment: "for better validation, I suggest using the calibrated parameter to reproduce a second flume experiment and discuss it."

**AUTHORS RESPONSE:** Thanks for Reviewer's good suggestion! As Reviewer suggested, multiple groups of validation simulations can indeed improve reliability of the numerical model, however, these are usually limited by the available data. In the Song et al. (2021) 's paper, only three groups of debris flow impact tests were published in detail, the test ID is 40-100-15, 50-100-15 and 55-100-15, respectively. There were two main reasons why the test ID of 50-100-15 was selected for model verification: (1) all the material remained quite well mixed with fluid phase, and a tiny separation between solid and fluid phase was visible during all the experiment, therefore, it was considered to be suitable for the RNG model validation. (2) the representative flip-through impact phenomenon was presented in this test (as added in Line 131-134 of the revised manuscript).

In the test group of 40-100-15, however, there was an obvious separation between solid and fluid phase when the flow was jetted vertically along the barrier face (see Fig. 5(a) and supplemental videos in Song et al. (2021)'s paper). In the test group of 55-100-15, static load by gradual filling of the subsequent debris was only demonstrated, due to the subcritical Froude condition, and the dynamic impact is not obvious (Song et al., 2021). It is obvious that the later two groups of experiments are not appropriate for the RNG model, and it is also proved by the poorer comparison results in terms of peak impact force, which were executed in our preliminary experiments.

**6. In the numerical modelling, the authors used a fixed discharge of 500 m³/s for a very short time (10 s). If the peak of discharge could be of some interest for very large debris**

flow, however, the duration and the constant value are not realistic and leads to unrealistic values of impact force. A more realistic debris flow inflow can be a triangular one where the overall duration is about 15 minutes with a peak discharge that occurs after 5 minutes (some examples of real and simplified hydrographs with can be found in Berger & al. 2011, Marchi & al. 2021). This modification in the inflow is essential for a truthful analysis of forces since, one of the main features of a debris flow just described previously, is the great deposition that occurs when the flow is slowed down. The direct consequence of the deposition is the time increase of the pressure due to this saturated terrain at rest.

**AUTHORS RESPONSE:** Thanks for Reviewer's comments! As Reviewer mentioned, the realistic debris flow inflow discharge and duration are essential for a truthful analysis of impact forces. A 3-D numerical simulation with a realistic hydrograph, however, needs a large number of computer memory and processing time (as added in Line 183-184 of the revised manuscript). In this study, the computer memory of a single simulation is about 11 GB under the computation time of 10 s (as added in Line 205 of the revised manuscript), however, it will jump to 948 GB in the time of 900 s (a realistic debris flow duration of 15 min). In term of processing time, a flood FLOW-3D simulation with the duration of 1020 s (17 min) took a maximum of 425 hours (Gems et al., 2016). It is much hardly acceptable for the sensitivity analysis, in which the sufficient experimental groups are required (as added in Line 184-185 of the revised manuscript).

Therefore, we set a fixed discharge of 500 $m^3$/s, which can be of some interest for a large magnitude of debris flow (as added in Line 186-187 of the revised manuscript). It is important to correct that the time of 10 s is not the inflow duration, but the computation time. The debris flow was ensured to flow out of the deposition fan and the target building was fully exposed during this time. It is important to emphasize that, therefore, the peak impact force involved in this study was the maximum value limited in the computation time of 10 s with a fixed discharge of 500 $m^3$ $s^{-1}$ (as added in Line 189-191 of the revised manuscript).

For the pressure induced by the deposition as Reviewer mentioned, we are very sorry that it was ignored due to the limits of the RNG model. It is demonstrated that the deposition of debris in front of the building increases the impact pressure indeed, resulting from the active earth pressure of the water-saturated sediment (Gao et al., 2017). However, it seems to has little effect on the peak impact force. From the laboratory results, this part of forces can be regarded as an almost constant static pressure at the end of the experiments, which is substantially lower than the peak impact force (Sturm et al., 2018a).

**7. Some perplexity will arise also by looking at some of the parameters used: roughness and viscosity. For the surface roughness, the authors used 0.05 m which represents "the equivalent grain roughness (or absolute height in meters)". This means that on all the surfaces of the computational domain (that also includes the buildings) the roughness is generated by grains of 5 centimetres. This kind of roughness can be representative of a natural environment (e.g., riverbed, grassland, wood) but in an urban environment, where usually the surfaces are paved or made of gravel, is too big. For the viscosity, the authors used 1 Pas. This value is at least one order of magnitude higher**

compared to the ones described by Iverson 1997 (the fluid viscosity ranges from 0.001 Pas to 0.1 Pas) or also the ones measured by Song & al. 2021 (laboratory experiment with fluid viscosity ranging from 0.001 up to 0.1). Also, the authors use a value of 0.1 Pas to validate the FLOW-3D model (based on one of the experiments of Song & al. 2021). Why this choice? If you validate the model with 0.1 Pas, also the other simulations should be performed with similar viscosity.

**AUTHORS RESPONSE:** Thanks for Reviewer's comments! We apologize for the incomplete statements about the surface roughness and fluid viscosity. For the surface roughness, the roughness parameter was only set for the deposition fan surface. As Review suggested, the surface roughness ($k$) of 0.05 m was set, meaning that the deposition fan surface is roughened with 5 cm diameter particles, for the representation of the natural environment surrounding mountainous buildings (as added in Line 178-180 of the revised manuscript).

For the fluid viscosity, we apologize for missing to explain the reason for setting the different viscosity in the validation and analysis models. As Reviewer mentioned, the fluid viscosity ranges indeed from 0.001 Pa·s to 0.1 Pa·s in Iverson et al. (1997) and Song et al. (2021) 's physical experiments. However, it doesn't mean that the viscosity of realistic debris flow slurry could only be limited in this range. After the model validation, the availability of the RNG and GMO coupled model in capturing the peak impact force and fluid surface was demonstrated, to a certain extent. Theoretically, this model can also be applied in the cases of the other viscosity, except for the viscosity of 0.1 Pa·s. In order to compare with some realistic building damage cases, for example the Qipan gully and Zhouqu debris flows in the west of China, the rheological properties of numerical model was set as the viscous debris flow. Therefore, the debris flow density was set as 2000 kg m-3 and viscosity was empirically 1.0 Pa·s (Fig. 2.3 in Takahashi, 2007) (as added in Line 192-194 of the revised manuscript).

**8. The target building has walls with a thickness of 0.35 m (line 156), while the cell (cube?) has a 0.25 m side (line 158). How is possible to simulate a wall that has a dimension that is not a multiple of a cell? Why not use a wall thickness equal to the cell side?**

**AUTHORS RESPONSE:** Thanks for Reviewer's comments! We apologize for neglecting to clarify the reasons for setting cell size of the target building domain. The statements about the refining principles for the embedded mesh were added in Line 197-198 and Line 199-204 of the revised manuscript.

**Line 197-198:**

To maintain a balance between the computational accuracy and time cost, the whole computation domain was discretized at intervals of 0.5 m.

**Line 199-204:**

The embedded domain of target building should be refined further following two principles: (1) its cell size must be less than the wall thickness of 0.35 m. This is because the computation grids can not rotate following the changing orientations of target building. Once the building rotating, some lacks of building surface will be produced, due to the half of a single cell can not

be covered by the wall element. (2) the boundaries of the embedded and external meshes must be overlapped for the computation stability, that is the external cell size is a multiple of embedded cell size. To sum up, the cell size of target building domain was determined as 0.25 m.

**9. The force is a vector, so it has an orientation. In the paper, I suppose, the authors report only its module. This aspect gives rise to two main questions. The first one is how the impact force is evaluated: is it evaluated also considering the tangential stresses on the walls? The second question is about where the force is evaluated: it is all over the surfaces of the building (inside and outside walls)? If the answer is yes, is it simply a sum of the force exerted by the mixture over all the walls? In this specific case, if there is flow inside the building, is the force on one wall the net force evaluated between inside and outside or is it the sum of the two? Moreover, is it considered also the roof?**

(1) Response to Question 1: "how the impact force is evaluated: is it evaluated also considering the tangential stresses on the walls?"

**AUTHORS RESPONSE:** We apologize for missing to describe the computation approach of impact force. The overall impact force of target building was gained from the combined fluid force considering the force direction in the GMO model, which was calculated from the normal pressures and shear forces in x, y and z directions (as added in Line 123-125 of the revised manuscript). Therefore, the tangential stresses on the walls have been considered in this study.

(2) Response to Question 2: "where the force is evaluated: it is all over the surfaces of the building (inside and outside walls)?..."

**AUTHORS RESPONSE:** Thanks for Reviewer's comments! All over the surfaces of target building are stressed elements, including the exterior wall, interior wall and roof (as added in Line 122-123 of the revised manuscript). The overall impact force is not simply the sum of absolute force values from each stressed element, but the combined force considering the force direction. In our preliminary experiment, the two sides of a wall was designed to be impacted simultaneously under the same hydrodynamic conditions. The gained impact force was not twice as the only one-side impact, on the contrary, it was too small to be ignored. As Reviewer assumed, in a specific case where the inside and outside wall are impacted simultaneously, the overall impact force is definitely the net force value between the impact forces of the two walls.

**10. When the azimuth angle A decreases and approaches 0, it has to be specified that the two surrounding buildings become a single building. Regarding this aspect, is the metamodel able to consider this? Otherwise, the authors have to be neglected, from the metamodel simulations, all the cases when the surrounding buildings are merged.**

**AUTHORS RESPONSE:** We greatly appreciate for Reviewer's comments! In this study, the metamodels were created from 160 simulation samples, in which the representative values of azimuth angle (**A**) were 0, 30, 45, 60 and 90, respectively. Only a single surrounding building

was set when the azimuthal angle was 0° (as described in Line 221-222 ). To a certain extent, therefore, the merging effect of surrounding buildings was considered in the metamodel modeling.

**11. Regarding the metamodel simulation, what are the ranges of variation of the four input variables?**

**AUTHORS RESPONSE:** Thanks for Reviewer's good suggestion! The statements about the variation ranges of the four input variables were added in Line 240-242 of the revised manuscript.

**Line 240-242:**

In the metamodel modeling, the variation range of the orientation (*Or*) was from 0° to 90°, the opening scale (*Op*) was from 0 to 0.8, the azimuthal angle (*A*) was 0° to 90° and the distance (*D*) was from 5 m to 30 m.

**12. I think that the duration of the simulation, which, from figures 17 and 21 it is set to 10 s (as the discharge duration), is too short since it for some tests the maximum value of the impact force is registered at the end of the simulation when a positive trend is also visible. I suggest increasing the simulation duration until the mixture is fully stopped or is flowed away from the target building.**

**AUTHORS RESPONSE:** Thanks for Reviewer's good suggestion! As responded in Question 6, It is important to emphasize that, therefore, the peak impact force involved in this study was the maximum value limited in the computation time of 10 s with a fixed discharge of 500 $m^3$ $s^{-1}$. During this time, the debris flow was ensured to flow out of the deposition fan and the target building was fully exposed.

[1] Franco, A., Moernaut, J., Schneider-Muntau, B., Strasser, M., Gems, B.: Triggers and consequences of landslide-induced impulse waves-3D dynamic reconstruction of the Taan Fiord 2015 tsunami event, Engineering Geology, 294, 106384, https://doi.org/10.1016/j.enggeo.2021.106384, 2021.

[2] Yin, Y. P., Huang, B., Chen, X., Liu, G., and Wang, S.: Numerical analysis on wave generated by the Qianjiangping landslide in Three Gorges Reservoir, China. Landslides, 12, 355-364, https://doi.org/10.1007/s10346-015-0564-7, 2015.

[3] Flow Science, Inc.: FLOW-3D v11.0.3 user manual, 2014.

[4] Song, D., Chen, X., Zhou, G. G. D., Lu, X., Cheng, G., Chen, Q.: Impact dynamics of debris flow against rigid obstacle in laboratory experiments, Engineering Geology, 291, 106211, https://doi.org/10.1016/j.enggeo.2021.106211, 2021.

[5] Gems, B., Mazzorana, B., Hofer, T., Sturm, M., Gabl, R., and Aufleger, M.: 3-D hydrodynamic modelling of flood impacts on a building and indoor flooding processes, Nat. Hazards Earth Syst. Sci., 16, 1351-1368, https://doi.org/10.5194/nhess-16-1351-2016, 2016.

[6] Gao, L., Zhang, L. M., and Chen, H. X.: Two-dimensional simulation of debris flow impact pressures on buildings, Engineering Geology, 226, 236-244, https://doi.org/10.1016/j.enggeo.2017.06.012, 2017.

[7] Sturm, M., Gems, B., Keller, F., Mazzorana, B., Fuchs, S., Papathoma-Köhle, M., and Aufleger, M.:

Understanding impact dynamics on buildings caused by fluviatile sediment transport, Geomorphology, 321, 45-59, https://doi.org/10.1016/j.geomorph.2018.08.016, 2018a.

[8] Iverson, R. M.: The physics of debris flows, Reviews of Geophysics. 35, 245–296, https://doi.org/10.1029/97RG00426, 1997.

[9] Takahashi, T.: Debris Flow Mechanics, Prediction and Countermeasures, Taylor & Francis Group, London, UK, 2007.

---

## Author Comment (AC2)

**RESPONSE TO REVIEWER COMMENTS:**

**Authors' General comment:** Thank you for the Reviewer's constructive comments concerning our manuscript entitled "Sensitivity analysis of a built environment exposed to debris flow impacts with 3-D numerical simulations" (ID: nhess-2022-173). Those comments are all valuable and very helpful for revising and improving our paper, as well as the important guiding significance to our researches. We have studied comments carefully and have made correction which we hope meet with approval. Revised portion are marked in red in the paper and response letter, and the manuscript is re-submitted in clean format to the Journal. Please also find below my response to Reviewer's comments.

**REVIEW COMMENTS:**

**According Martinez-Carvajal et al (2018), a natural phenomenon (hazard) may be characterized in terms of temporal, spatial and magnitude probabilities. The effects of the interaction between the hazard and the exposed element depend on the intensity of the hazard and on the resistance, sometimes called susceptibility, of the element at risk, which describes the propensity of a building or other infrastructure to suffer damage from a specific hazard impact. Consequently, a modern concept of vulnerability must consider the intensity of the hazard as well as the structural resistance of the exposed infrastructure. This concept is referred to as physical vulnerability, and the most accepted definition is a representation of the expected degree of loss quantified on a scale of 0 (no damage) to 1 (total destruction).**

**Previous considerations leads me to suggest to the authors the inclusion of a broad discussion on vulnerability which certainly is the major objective of this kind of research. Comments on the effect of the buildings strength will be profitable for opening future research topics by means of numerical modelling.**

**AUTHORS RESPONSE:** We greatly appreciate for Reviewer's good comments! As Reviewer suggested, a broader discussion on the contributions of the present paper towards debris flow hazard and vulnerability assessment was added in Line 546-568 of the revised manuscript.

**Line 546-568:**

It is obvious that the quantitative descriptions about the interactions between the built environment and impact forces can be useful to the built environment improvement and local adaptation measures for the impact force reduction, which are assumed as the low-cost and efficient approach for mitigating the building's structural damages. And more significantly, the present paper has extended the knowledge about the influence factors on debris flow intensity. It is demonstrated that some artificial building factors can not be ignored, except for the natural environments, in deciding the spatial pattern of the process intensity. The further research about their relative importance with the 3-D numerical simulation and sensitivity analysis can promote the relative intensity evaluation of the building, especially in terms of the indicator selection and weighting, which may open a future topic of the debris flow hazard assessment. For the building vulnerability assessment, the indicators can be mainly divided into two kinds:

the exterior process intensity and interior building resistance. The process intensity, for example the flow depth, velocity, impact force or the other proxy, was assumed absolutely necessary, either in the curve based approach or the indicator based approach (Martinez-Carvajal et al., 2018). From the current literature, however, there are some confusions in selecting the surroundings factors and process intensity indicator. To be specific, some surroundings factors or also called protection factors, including the *Surrounding buildings*, *Building row*, *Wall around building*, *Natural barriers* and so on, were still selected when the debris flow intensity had been indirectly considered (Dall'Osso et al., 2009; Dall'Osso et al., 2016; Papathoma-Köhle et al., 2019). These indicators should be independent each other theoretically. From the views of the present paper, the functions of all over the surroundings factors are the influences on the process intensity around building. Therefore, the process intensity should be exclusive with the surroundings factors. The building features factors are mainly considered to be acted on the building resistance, including the *Material*, *Structure*, *Number of stories*, *Foundation strength* and so on. However, It is not hard to find that some building indicators, for example the *Orientation*, *Shape* and *Openings*, can rebuild the process intensity. As a result, the effect of the representative building features indicators on the building vulnerability needs an in-depth discussion in future. The last but not least, a more universal, robust index may be developed using the numerical simulation approach, which can improve the locality limits resulting from the empirical data, to some extent.

[1] Martinez-Carvajal, H. E., de Moraes Guimaraes Silva, M. T., Garcia-Aristizabal, E. F., Aristizabal-Giraldo, E. V., Larios-Benavides, M. A.: A mathematical approach for assessing landslide vulnerability. Earth Sciences Research Journal, 22, 251-273. https://doi.org/10.15446/esrj.v22n4.68553, 2018.

[2] Dall'Osso, F., Gonella, M., Gabbianelli, G., Withycombe, G., Dominey-Howes, D.: A revised (PTVA) model for assessing the vulnerability of buildings to tsunami damage, Nat. Hazards Earth Syst. Sci., 9, 1557-1565, https://doi.org/10.5194/NHESS-9-1557-2009, 2009.

[3] Dall'Osso, F., Dominey-Howes, D., Tarbotton, C., Summerhayes, S., Withycombe, G.: Revision and improvement of the PTVA-3 model for assessing tsunami building vulnerability using "international expert judgment": introducing the PTVA-4 model, Nat Hazards, 83, 1229-1256, https://doi.org/10.1007/s11069-016-2387-9, 2016.

[4] Papathoma-Köhle, M., Schlögl, M., and Fuchs, S.: Vulnerability indicators for natural hazards: an innovative selection and weighting approach, Scientific Reports, 9, 15026, https://doi.org/10.1038/s41598-019-50257-2, 2019.

---

## Author Comment (AC3)

**RESPONSE TO REVIEWER COMMENTS:**

**Authors' General comment:** Thank you again for the Reviewer's constructive comments concerning our manuscript entitled "Sensitivity analysis of a built environment exposed to debris flow impacts with 3-D numerical simulations" (ID: nhess-2022-173). We have the greatest respect for Reviewer's professional opinions on the debris flow hazard. Reviewer's comments are all valuable and very helpful for revising manuscript and improve our research. We have studied comments carefully and have made corrections which we hope meet with approval. Revised portions are marked in red in the paper and response letter, and the manuscript is re-submitted in clean format to the Journal. Please also find below my response to Reviewer's comments.

**REVIEW COMMENTS:**

**1. The main one concern the validation. In the laboratory experiment used for the validation, a well-mixed stony granular debris flow is reproduced. One of the main characteristics of this debris flow is that the energy dissipation is due to the collision between the particles and not by the viscosity of the fluid (e.g. Iverson 1997, Takahashi 2007, Armanini 2013). However, in the authors' response to my comment #1, it is highlighted that "From the characteristics of RNG k-ε model [that is used in all the manuscript], the type of debris flow involved in this study was determined as mudflow or viscous debris flow, in which a singlephase non-Newtonian fluid was assumed and solid particles were treated as suspension and mixed with the fluid phase well". This statement is completely in contrast with the used laboratory experiment used and consequently, all the section devoted to the validation of the model is meaningless since the author used a model that could not represent correctly the physical processes involved.**

**AUTHORS RESPONSE:** We greatly appreciate for Reviewer's good comments! As Reviewer suggested, we have selected another classic dam-break experiment for the validation of fluid-structure interaction (Gomez-Gesteira and Dalrymple, 2004; Liu et al., 2021). The new statements about the model validation were added in Line 128-157 of the revised manuscript.

**Line 128-157:**

**2.2 Model validation**

The interaction between a dam-break and the structure has become a classic benchmark for the validation of fluid-structure interaction (Liu et al., 2021). The accuracy of the model will be validated by means of the experimental setup previously used in Gomez-Gesteira and Dalrymple (2004). This experiment has been referred as a "bore in a box", where it was a dam-break and structure-impact problem confined within a rectangular box. The geometric dimensions of the experimental model are shown in Fig. 1.The rectangular tank is 1.60 m long, 0.61 m wide and 0.75 m high.The volume of water initially contained behind a thin gate at one end of the box is 0.4 m long, 0.61 m wide and 0.3 m high. An initial layer of water (approximately 1 cm deep) existed on the bottom of the tank. The obstacle, which is 0.12 m × 0.12 m × 0.75 m in size, is placed 0.5 m downstream of the gate and 0.24 m from the nearest sidewall of the tank. The time history of the impact force on the structure was measured with a load cell.

In the numerical simulation, the analysis domain was discretized into a grid with a cell size of 0.01 m, which was equal to a cube of 0.01 m side in 3-D model. The fluid properties were set to be the density of 1000 kg m$^{-3}$ and viscosity of 0.001 Pa·s. The motion of fluid was computed by means of RNG k-ε model in FLOW-3D. The obstacle and gate were controlled by the GMO module, specifically this obstacle was set as a fixed and non-deformable rigid body, and the gate was prescribed to be lifted 0.3 m along the $Z^+$ direction. The time history of impact forces and the corresponding dynamic processes were selected to validate the accuracy of the numerical simulation. The direction of the force was considered positive when exerted in the $Y^+$ direction.

Fig. 2a shows the agreement of numerical forces obtained by means of the RNG and GMO coupled model with experimental data, particularly the positions of both peaks, which correspond to the wave hitting the front and the back of the structure and were reproduced perfectly by the numerical model. Fig. 2b shows the evolution of the wave generated by the dam-break and the initial layer of water on the bottom. At t = 0.32 s, the wave is colliding with the front of the obstacle. At t = 0.58 s, the wave is wrapping around the structure, colliding together and continues moving toward the tank wall. At t = 1.44 s, the reflected wave is hitting the back of the obstacle.

[Figure]

**Figure 1. The geometric dimensions of the experimental dam-break model**

[Figure]

**Figure 2.** The dam-break simulation (a) comparison between numerical (red line) and experimental values (blue line) of the force exerted on the structure; (b) wave evolution (t = 0.32 s) the wave colliding with the front of the obstacle; (t = 0.58 s) the wave wrapping around the structure, colliding together and continues moving toward the tank wall; (t = 1.44 s) the reflected wave hitting the back of the obstacle.

**2. Another point of the validation part regards why the authors do not show the time history of the impact force. Since one of the characteristics of a debris flow impact process is its dynamic changes in time as the experiments of Song et al. 2021 show (the time history is quite complex and is not only represented by a single value!), the "simple" peak value is not sufficient for validating the model used. For this reason, I think that the authors' response "It is demonstrated that the RNG and GMO coupled model in FLOW-3D are able to describe the peak impact force and fluid surface effectively" is not fully trustable.**

**AUTHORS RESPONSE:** Thanks for Reviewer's good suggestion! As responded in the comment #1, a new model validation has been executed, and the time history of impact force was determined to be compared, as shown in Fig. 2a.

**3. A third comment regards the author's answer to comment #6 in combination with #12. I know well that long simulations use a high quantity of memory and take long computational times, so for this reason it could be, in some cases, acceptable to use high fixed discharge for a short time. However, I think that the 10-second duration used by the authors is not fully appropriate at least for some of the simulations used. For example, it is clear from Figure 17 that for the simulation with 45° of orientation (Or45) the peak impact force is the last value of the plot (i.e. at 10 second, so at the end of the simulation) but the force has a trend is still increasing! Also for the cases of 60° and 30° (i.e Or60 and Or30), the trend of the force is still increasing and at the end of the**

simulation (i.e. the end of the plot), the values are very close to the peak values. This means that, at least, in these three simulations (but I think that the same problem arises also in lots of other simulations done by the authors as the one shown in Figure 21) the authors have to increase the time of the simulation until a significant (a few seconds?) decreasing, or at least constant, value of the impact force is visible.

**AUTHORS RESPONSE:** Thank you very much! We apologize for Reviewer's confusion induced by our unclear responses. In this study, the time of 10 s was not referred to the duration of debris flow hydrograph, but the computation time in FLOW-3D, that was 10 s after the debris flow was released from the inflow point. During this time, the debris flow head was ensured to move to the edge of deposition fan and the target building was fully exposed. It is important to emphasize that, therefore, the peak impact force involved in this study was the maximum value limited in the computation time of 10 s under a fixed discharge of 500 $m^3$ $s^{-1}$ (as added in Line 170-174 of the revised manuscript). As Reviewer mentioned, the peak impact force maybe not the maximum within a whole hydrograph. This is a very good suggestion, the longer simulation time will be taken into consideration in the future research.

**4. The last comment regards the author's response to comment #9 regarding the force. For me, it remains unclear the meaning of the number that represents the force. Moreover, since the target building is a complex geometry, where these "numbers" are applied? It is quite different if this "number" is applied only to a single surface (e.g. only on the wall with the opening) or it is applied with different values on different surfaces (e.g. on the wall with the opening plus the roof) because the possible consequences are completely different! Here, I speak about "number" since, as in comment #9, I underline that the force is a vector, so a simple "number" does not represent the force: it is still missing the direction of this force and the point where the force is applied!.**

**AUTHORS RESPONSE:** Thank you very much! We apologize for Reviewer's confusion induced by our unclear responses. Reviewer's comment on the overall impact force of target building can be concluded as two questions, specifically the meaning of the force number and position of force application. On the first question, the impact force is the combined hydraulic force due to the normal pressure and shear stress in the space system. With the help of GMO model in FLOW-3D, the normal pressure and shear force of the impacted object in $X$, $Y$ and $Z$ directions can be calculated at each time step (as added in Line 118-120 of the revised manuscript). On the second question, due to the complex geometry and variable built environments, the impacted elements of target building are changing in the different scenarios. For example, the wall B was impacted mainly in the orientation of 0° (Fig. 15a), and the wall A was impacted mainly in the Or90 scenario (Fig. 15e). In this study, therefore, the target building was treated as a whole bearing structure to keep consistency of analysis, that was the structural components, including the column, beam and bearing wall, were not analyzed separately. All over the grids covering building surface would be calculated when contacting with the flow (as added in Line 120-123 of the revised manuscript). As above-mentioned, the overall impact force is referred to the magnitude of combined fluid force, which is calculated from the all meshes covering the target building surface when impacted with the fluid. As Reviewer suggested, however, it is quite different damage state when the same magnitude of impact force is applied on the different position. This is a very good suggestion, we will take it into consideration in-depth in the future research.

[1] Gomez-Gesteira, M., Dalrymple, R. A.: Using a three-dimensional smoothed particle hydrodynamics method for wave impact on a tall structure. J Waterw Port Coast, 130, 63-69, https://doi.org/10.1061/(ASCE)0733-950X(2004)130:2(63), 2004.

[2] Liu, C., Yu, Z., and Zhao, S.: A coupled SPH-DEM-FEM model for fluid-particle-structure interaction and a case study of Wenjia gully debris flow impact estimation, Landslides, 18, 2403-2425, https://doi.org/10.1007/s10346-021-01640-6, 2021.

---

## Referee Report (RR1)

**Review of manuscript NHESS-2022-173 – revised version**

Sensitivity analysis of a built environment exposed to debris flow impacts with 3-D numerical simulations.

**Overview**

I appreciate how the authors modify the manuscript and try to answer all the questions proposed by the Reviewers in their comments. However, some questions remain still open while others arise from the revised version of the paper.

**Observations and questions**

In the following, all the questions and observations arose from the reading of the new version of the manuscript.

1. I know very well that the longer the simulation, the longer the time used and the bigger the amount of f data to be analysed, but I think it is not entirely acceptable that all the analysis is developed with a limited time for the evaluation of the maximum peak impact force. As mentioned in [RC1] and [RC3] for all the two cases in which the time variation of the impact force is reported (so Figure 17 and Figure 21) the force trend is still increasing at the end of the plot (simulation) and some maximum values are evaluated exactly at the end. Additionally, some intersections between the impact force are present depending on the variable analysed. This means that if the simulation is longer than the used 10s and since the maximum values can increase, it can happen that also the analysis of the variable involved can change giving rise to a modification of the effect: by way of example only, it can happen that if the simulation is longer, the peak impact force of the case Or30-Op0-Anull-Dnull becomes greater than Or0-Op0-Anull-Dnull so all the discussion provided by the author present a limited scientific meaning. This means that lines 172-174 were not sufficient as an explanation for the time length, nor the statement that a longer simulation requires too much time).
I suggest that in at least one of the tests where the time variability of the impact force is plotted, the authors produce longer simulation (for example, as suggested in [RC3] "*increase the time of the simulation until a significant (a few seconds?) decreasing, or at least constant, value of the impact force is visible*") in order to show that the used value produce something that has real scientific relevance. I think that providing this for at least one involved variable will demonstrate that the peak impact force values used are trustworthy.
2. The title can be misleading. The paper tackle monophasic viscous debris flow with a (very) synthetic and simplified hydrograph. I think that the underline words, or their meaning, must appear in the title. These concepts must appear also in the Abstract.
3. Line 12: "peak impact forces" -> It is not completely true since is the value obtained by a limited-time simulation of 10s.
4. Line 105: "transport equations" -> It is formally correct to say that equations (1) and (2) are the transport equation of the RNG k-ε model. However, some readers used to deal with sediment moving over the fixed bed and/or flow tracers (e.g., chemical components advected by a flow field) can be confused. I think it is better to say that the equations describe the turbulent kinetic energy ($k_T$) and the turbulence dissipation ($\varepsilon_T$) balance equations of the RNG k-ε model
5. Line 147 "perfectly" -> If the numerical model perfectly reproduces something, no difference with experimental results must be present and this is not the case. Please modify.
6. Lines 162-164 -> The authors describe the values used for the roughness of the inclined plane where the viscus debris flow moves, but what about the building? Are they without roughness, so are they treated

as smooth surfaces? Please justify the choice. Moreover, it is missing the roughness used in the model validation.

7. Lines 167-174 -> From the text it seems that the discharge, that feeds the computational domain, is considered constant and equal to 500 m³/s for all the duration of the simulations (the authors wrote on lines 168-169 "*an inflow with a time-invariance discharge of 500 m³ s⁻¹*" and on lines 173-174 "*in the computation time of 10 s under a fixed discharge of 500 m³ s⁻¹*"). However, in the response to the reviewer, the authors highlighted that "*In this study, the time of 10 s was not referred to the duration of debris flow hydrograph, but the computation time in FLOW-3D, that was 10 s after the debris flow was released from the inflow point*" that seems something completely different from the previous statement. From this answer, I understand that the simulation time is 10 s, but the discharge of 500 m³ s⁻¹ has a duration completely different and it seems also that this discharge occurs only for a few time steps (or only in the first one). Please clarify this aspect because it is fundamental for understanding the work: the general meaning of the work is completely different if the discharge lasts for 10s or for only some time steps.

8. Lines 182-187 -> For me it is not fully clear how the wall dimension can be 0.35m. If I understand well the text, the domain associated with the building is discretized with cells of 0.25m (that is half the size of the cell used in the rest of the computational domain). But if the dimension of the building cells is 0.25m, how is it possible to have a wall dimension of 0.35m? Is the 0.35m value an average value between the different configurations (I understand that the cell could not rotate, so in certain cases, the wall is discretized by, for example, 2 cells)? Or is there present some numerical procedure that can subdivide the building cells into fractions? Please clarify this point.

---

## Author Response (AR2)

**RESPONSE TO REVIEWER COMMENTS:**

**Authors' General comment:** Thank you again for the Reviewer's constructive comments concerning our manuscript entitled "Sensitivity analysis of a built environment exposed to debris flow impacts with 3-D numerical simulations" (ID: nhess-2022-173). We have the greatest respect for Reviewer's professional opinions on the debris flow hazard. Reviewer's comments are all valuable and very helpful for revising manuscript and improve our research. We have studied comments carefully and have made corrections which we hope meet with approval. Revised portions are marked in red in the paper and response letter, please find below my response to Reviewer's comments.

**REVIEW COMMENTS:**

**1. I know very well that the longer the simulation, the longer the time used and the bigger the amount of data to be analysed, but I think it is not entirely acceptable that all the analysis is developed with a limited time for the evaluation of the maximum peak impact force. As mentioned in [RC1] and [RC3] for all the two cases in which the time variation of the impact force is reported (so Figure 17 and Figure 21) the force trend is still increasing at the end of the plot (simulation) and some maximum values are evaluated exactly at the end. Additionally, some intersections between the impact force are present depending on the variable analysed. This means that if the simulation is longer than the used 10s and since the maximum values can increase, it can happen that also the analysis of the variable involved can change giving rise to a modification of the effect: by way of example only, it can happen that if the simulation is longer, the peak impact force of the case Or30-Op0-Anull-Dnull becomes greater than Or0-Op0-Anull-Dnull so all the discussion provided by the author present a limited scientific meaning. This means that lines 172-174 were not sufficient as an explanation for the time length, nor the statement that a longer simulation requires too much time.**

**I suggest that in at least one of the tests where the time variability of the impact force is plotted, the authors produce longer simulation (for example, as suggested in [RC3] "increase the time of the simulation until a significant (a few seconds?) decreasing, or at least constant, value of the impact force is visible") in order to show that the used value produce something that has real scientific relevance. I think that providing this for at least one involved variable will demonstrate that the peak impact force values used are trustworthy.**

**AUTHORS RESPONSE:** We greatly appreciate for Reviewer's good comments! In the previous simulations, the debris flow discharge at the inflow cross section was set to be fixed (500 $m^3$ $s^{-1}$), and the duration of hydrograph was not limited. The debris flow, in other words, was released endlessly from the inflow cross section. The previous computation time of FLOW-3D model was set 10 s, and the peak impact force was considered as the maximum value in this time range. As Reviewer suggested, however, it's hardly to keep the impact process stable. It is found that some peak impact force, therefore, was evaluated at the end of simulation, when the force trend was still increasing. In order to address the problem completely, **the duration of debris flow hydrograph and computation time of FLOW-3D model for all the simulation scenarios are reconsidered in the revised manuscript.** Specifically, the debris flow discharge is still fixed as 500 $m^3$ $s^{-1}$, and the duration time is set 5 s at the inflow cross section. In consideration of the distance between inflow cross section and target building, the entire simulation

computation time is extended to 15 s to ensure that nearly all the debris flow can flow through the target building (as described in Line 176-182 of the revised manuscript). During this time, there are the distinct rise and fall trends in the impact process line, to ensure the peak impact force is evaluated more scientifically. And all the relevant simulation data and analysis results have been updated in the revised manuscript (as marked in red).

**2. The title can be misleading. The paper tackle monophasic viscous debris flow with a (very) synthetic and simplified hydrograph. I think that the underline words, or their meaning, must appear in the title. These concepts must appear also in the Abstract.**

**AUTHORS RESPONSE:** Thanks for Reviewer's good suggestion! As Reviewer suggested, the title has been updated as "*Sensitivity analysis of a built environment exposed to a synthetic monophasic viscous debris flow impacts with 3-D numerical simulations*". And the Abstract has also been revised accordingly, as added in Line 15-16 of the revised manuscript.

**Line 15-16:**

The impact forces were obtained from the monophasic viscous debris flow with a synthetic and simplified hydrograph using the FLOW-3D model...

**3. Line 12: "peak impact forces" -> It is not completely true since is the value obtained by a limited-time simulation of 10s.**

**AUTHORS RESPONSE:** We greatly appreciate for Reviewer's good comments! As described in Question 1#, the modified computation time has been extended to be triple of the inflow duration, in order to ensure nearly all the debris flow can flow through the target building. The peak impact force, therefore, is treated as the true maximum value in a relatively complete impact process (as added in Line 182-183 of the revised manuscript).

**4. Line 105: "transport equations" -> It is formally correct to say that equations (1) and (2) are the transport equation of the RNG k-ε model. However, some readers used to deal with sediment moving over the fixed bed and/or flow tracers (e.g., chemical components advected by a flow field) can be confused. I think it is better to say that the equations describe the turbulent kinetic energy ($k_T$) and the turbulence dissipation ($\varepsilon_T$) balance equations of the RNG k-ε model.**

**AUTHORS RESPONSE:** Thanks for Reviewer's good suggestion! As Reviewer suggested, the statement of "transport equations" has been revised as "turbulent kinetic energy and the turbulence dissipation balance equations" (as added in Line 108 of the revised manuscript).

**5. Line 147 "perfectly" -> If the numerical model perfectly reproduces something, no difference with experimental results must be present and this is not the case. Please modify.**

**AUTHORS RESPONSE:** Thanks for Reviewer's good suggestion! As Reviewer suggested, the statement

of "perfectly reproduced" is indeed too absolute. This statement is modified in Line 152-154 of the revised manuscript.

**Line 146-148:**

Fig. 2a shows the general agreement of numerical forces obtained by means of the RNG and GMO coupled model with experimental data, particularly the positions of both peaks, which correspond to the wave hitting the front and the back of the structure and were reasonably reproduced by the numerical model.

**6. Lines 162-164 -> The authors describe the values used for the roughness of the inclined plane where the viscus debris flow moves, but what about the building? Are they without roughness, so are they treated as smooth surfaces? Please justify the choice. Moreover, it is missing the roughness used in the model validation.**

**AUTHORS RESPONSE:** We greatly appreciate for Reviewer's good comments! Firstly, the surface roughness of the obstacle is not determined in the physical test. Therefore, $k_o$ (0, 0.001, 0.002, or 0.003 m) is selected for sensitivity analysis to determine if this parameter could reasonably reflect the macro mechanical behaviors of the dam-break test (as described in Line 141-143 of the revised manuscript). It is indicated that the impact force is very low sensitivity to $k_o$ in Fig.2a, this study does not pay too much attention to the value of this parameter, and the surface roughness ($k$) of all the impacted object in numerical simulation is set 0 m (as described in Line 154-156 of the revised manuscript). Secondly, the tank is considered to be smooth surface in the model validation, and its surface roughness ($k$) is set 0 m (as added in Line 137-138 of the revised manuscript).

[Figure]

**Figure 2a. The dam-break simulation comparison between numerical (dotted lines) and experimental values (red line) of the force exerted on the structure.**

**7. Lines 167-174 -> From the text it seems that the discharge, that feeds the computational domain, is considered constant and equal to 500 m³/s for all the duration of the simulations (the authors wrote on lines 168-169 "an inflow with a time-invariance discharge of 500 m³s⁻¹" and on lines**

**173-174 "in the computation time of 10 s under a fixed discharge of 500 m³s⁻¹").** However, in the response to the reviewer, the authors highlighted that "In this study, the time of 10 s was not referred to the duration of debris flow hydrograph, but the computation time in FLOW-3D, that was 10 s after the debris flow was released from the inflow point" that seems something completely different from the previous statement. From this answer, I understand that the simulation time is 10 s, but the discharge of 500 m³s⁻¹ has a duration completely different and it seems also that this discharge occurs only for a few time steps (or only in the first one). Please clarify this aspect because it is fundamental for understanding the work: the general meaning of the work is completely different if the discharge lasts for 10s or for only some time steps.

**AUTHORS RESPONSE:** We greatly appreciate for Reviewer's good comments! We apologize for Reviewer's confusion induced by our unclear statements. In the newly revised manuscript, as described in Question 1#, the duration of 500 m³ s⁻¹ discharge is adjusted as 5 s at the inflow cross section. In consideration of the distance between inflow cross section and target building, the entire computation time is extended to 15 s, the triple of inflow duration, to ensure that nearly all the debris flow can flow through the target building. During this time, there are the distinct rise and fall trend in the impact force time history line, and the peak impact force is treated as the true maximum value in a relatively complete impact process (as added in Line 176-183 of the revised manuscript).

**8. Lines 182-187 -> For me it is not fully clear how the wall dimension can be 0.35m. If I understand well the text, the domain associated with the building is discretized with cells of 0.25m (that is half the size of the cell used in the rest of the computational domain). But if the dimension of the building cells is 0.25m, how is it possible to have a wall dimension of 0.35m? Is the 0.35m value an average value between the different configurations (I understand that the cell could not rotate, so in certain cases, the wall is discretized by, for example, 2 cells)? Or is there present some numerical procedure that can subdivide the building cells into fractions? Please clarify this point.**

**AUTHORS RESPONSE:** We thank Reviewer very much for the comments! Yes! Just as Reviewer assumed, FLOW-3D code can subdivide the building cells into fractions. The FLOW-3D procedure uses an unique FAVOR™ technique, an acronym for *Fractional Area/Volume Obstacle Representation*, to computes the open area fractions (AFT, AFR, AFB) on the cell faces along with the open volume fraction (VF) for reconstructing the geometry for a simulation. This approach offers a simple and accurate way to represent complex surfaces in the domain without requiring a body-fitted grid. The FAVOR processor can generate area fractions for each cell face in the grid by determining which corners of the face are inside of a defined geometry, and incorporate geometry effects into the governing equations (as added in Line 96-97 of the revised manuscript). If all four corners of a cell face are inside the geometry, then the entire face is defined to be within the geometry (as illustrated the red cell in Fig.8-1). Similarly, if all corners lie outside, then the entire face is assumed to be outside the geometry (as illustrated the circle in the lower right corner of the mesh). When some face corners are inside a geometry and some are outside, the intersection of the geometry with face edges are computed. Area fractions are then computed from these intersection points assuming straight-line connections between intersection points within the face (as illustrated the blue cell).

However, for some geometries and mesh resolutions it is possible that the geometry may intersect a cell face more than once. In this case the corresponding cell edge is assumed to be either fully inside the

object or fully outside (as illustrated the orange cell). This may happen in case of building rotating of the present study, as illustrated in Fig.8-2, the wall geometry can not be captured accurately (as added in Line 191-193 of the revised manuscript). The representation is improved as the mesh resolution is increased (i.e., the cell size is decreased). When the cell size is decreased from 0.35 m to 0.25 m, as shown in Fig.8-3, the geometry recognition accuracy has been greatly improved.

[Figure]

Figure 8-1. Object definition (left) and object created (right)

[Figure]

Figure 8-2. Object definition (left) and object created (right)
(The red wall is rotated counterclockwise by 30°, the wall thickness is 0.35 m, the blue cell size is 0.35 m)

[Figure]

Figure 8-3. Object definition (left) and object created (right)

(The red wall is rotated counterclockwise by 30°, the wall thickness is 0.35 m, the blue cell size is 0.25 m)

---

## Author Response (AR3)

**RESPONSE TO REVIEWER COMMENTS:**

**Authors' General comment:** Thank you again for the Reviewer's constructive comments concerning the manuscript entitled "Sensitivity analysis of a built environment exposed to the synthetic monophasic viscous debris flow impacts with 3-D numerical simulations" (ID: nhess-2022-173). Reviewer's comments are all valuable and very helpful for revising manuscript and improve our research. We have studied comments carefully and have made corrections which we hope meet with approval.

**REVIEW COMMENTS:**

**1. Figure 17-> delete "The time of peak impact force are marked with dot." since no dots are reported.**

**AUTHORS RESPONSE:** Thank you very much! This statement has been deleted from the illustrations in Fig.17.

**2. Figure 20 -> delete "The time of peak impact force are marked with dot." since no dots are reported.**

**AUTHORS RESPONSE:** Thank you very much! This statement was present in Fig.21, not Fig.20. We have deleted this sentence from the illustrations of Fig.21.

**3. Moreover, use the same velocity range for the three plots in Figure 2(b). In this way is more simple to compare how the velocity magnitude changes over time.**

**AUTHORS RESPONSE:** We greatly appreciate for Reviewer's good comments! As Reviewer suggested, the velocity range of three plots in Figure 2(b) has been set to be same, as shown below.

[Figure]